# Moser Flow: Divergence-based Generative Modeling on Manifolds

**Noam Rozen**[1]    **Aditya Grover**[2,3]    **Maximilian Nickel**[2]    **Yaron Lipman**[1,2]

[1]Weizmann Institute of Science    [2]Facebook AI Research    [3]UCLA

## Abstract

We are interested in learning generative models for complex geometries described via manifolds, such as spheres, tori, and other implicit surfaces. Current extensions of existing (Euclidean) generative models are restricted to specific geometries and typically suffer from high computational costs. We introduce *Moser Flow* (MF), a new class of generative models within the family of continuous normalizing flows (CNF). MF also produces a CNF via a solution to the change-of-variable formula, however differently from other CNF methods, its model (learned) density is parameterized as the source (prior) density minus the *divergence* of a neural network (NN). The divergence is a local, linear differential operator, easy to approximate and calculate on manifolds. Therefore, unlike other CNFs, MF does not require invoking or backpropagating through an ODE solver during training. Furthermore, representing the model density explicitly as the divergence of a NN rather than as a solution of an ODE facilitates learning high fidelity densities. Theoretically, we prove that MF constitutes a universal density approximator under suitable assumptions. Empirically, we demonstrate for the first time the use of flow models for sampling from general curved surfaces and achieve significant improvements in density estimation, sample quality, and training complexity over existing CNFs on challenging synthetic geometries and real-world benchmarks from the earth and climate sciences.

## 1   Introduction

The major successes of deep generative models in recent years are primarily in domains involving Euclidean data, such as images (Dhariwal and Nichol, 2021), text (Brown et al., 2020), and video (Kumar et al., 2019). However, many kinds of scientific data in the real world lie in non-Euclidean spaces specified as manifolds. Examples include planetary-scale data for earth and climate sciences (Mathieu and Nickel, 2020), protein interactions and brain imaging data for life sciences (Gerber et al., 2010; Chen et al., 2012), as well as 3D shapes in computer graphics (Hoppe et al., 1992; Kazhdan et al., 2006). Existing (Euclidean) generative models cannot be effectively applied in these scenarios as they would tend to assign some probability mass to areas outside the natural geometry of these domains.

An effective way to impose geometric domain constraints for deep generative modeling is to design *normalizing flows* that operate in the desired manifold space. A normalizing flow maps a prior (source) distribution to a target distribution via the change of variables formula (Rezende and Mohamed, 2015; Dinh et al., 2016; Papamakarios et al., 2019). Early work in this direction proposed invertible architectures for learning probability distributions directly over the specific manifolds defined over spheres and tori (Rezende et al., 2020). Recently, Mathieu and Nickel (2020) proposed to extend continuous normalizing flows (CNF) (Chen et al., 2018) for generative modeling over Riemannian manifolds wherein the flows are defined via vector fields on manifolds and computed as the solution to an associated ordinary differential equation (ODE). CNFs have the advantage that the neural

network architectures parameterizing the flow need not be restricted via invertibility constraints. However, as we show in our experiments, existing CNFs such as FFJORD (Grathwohl et al., 2018) and Riemannian CNFs (Mathieu and Nickel, 2020) can be slow to converge and the generated samples can be inferior in capturing the details of high fidelity data densities. Moreover, it is a real challenge to apply Riemannian CNFs to complex geometries such as general curved surfaces.

To address these challenges, we propose Moser Flows (MF), a new class of deep generative models within the CNF family. An MF models the desired target density as the source density minus the divergence of an (unrestricted) neural network. The divergence is a local, linear differential operator, easy to approximate and calculate on manifolds. By drawing on classic results in differential geometry by Moser (1965) and Dacorogna and Moser (1990), we can show that this parameterization induces a CNF solution to the change-of-variables formula specified via an ODE. Since MFs directly parameterize the model density using the divergence, unlike other CNF methods, we do not require to explicitly solve the ODE for maximum likelihood training. At test-time, we use the ODE solver for generation. We derive extensions to MFs for Euclidean submanifolds that efficiently parameterize vector fields projected to the desired manifold domain. Theoretically, we prove that Moser Flows are universal generative models over Euclidean submanifolds. That is, given a Euclidean submanifold $\mathcal{M}$ and a target continuous positive probability density $\mu$ over $\mathcal{M}$, MFs can push arbitrary positive source density $\nu$ over $\mathcal{M}$ to densities $\bar{\mu}$ that are arbitrarily close to $\mu$.

We evaluate Moser Flows on a wide range of challenging real and synthetic problems defined over many different domains. On synthetic problems, we demonstrate improvements in convergence speed for attaining a desired level of details in generation quality. We then experiment with two kinds of complex geometries. First, we show significant improvements of $49\%$ on average over Riemannian CNFs (Mathieu and Nickel, 2020) for density estimation as well as high-fidelity generation on 4 earth and climate science datasets corresponding to global locations of volcano eruptions, earthquakes, floods, and wildfires on spherical geometries. Next and last, we go beyond spherical geometries to demonstrate for the first time, generative models on general curved surfaces.

## 2 Preliminaries

**Riemannian manifolds.** We consider an orientable, compact, boundaryless, connected $n$-dimensional *Riemannian manifold* $\mathcal{M}$ with metric $g$. We denote points in the manifold by $x, y \in \mathcal{M}$. At every point $x \in \mathcal{M}$, $T_x\mathcal{M}$ is an $n$-dimensional tangent plane to $\mathcal{M}$. The metric $g$ prescribes an inner product on each tangent space; for $v, u \in T_x\mathcal{M}$, their inner product w.r.t. $g$ is denoted by $\langle v, u \rangle_g$. $\mathfrak{X}(\mathcal{M})$ is the space of smooth (tangent) vector fields to $\mathcal{M}$; that is, if $u \in \mathfrak{X}(\mathcal{M})$ then $u(x) \in T_x\mathcal{M}$, for all $x \in \mathcal{M}$, and if $u$ written in local coordinates it consists of smooth functions. We denote by $dV$ the *Riemannian volume form*, defined by the metric $g$ over the manifold $\mathcal{M}$. In particular, $V(\mathcal{A}) = \int_{\mathcal{A}} dV$ is the volume of the set $\mathcal{A} \subset \mathcal{M}$.

We consider *probability measures* over $\mathcal{M}$ that are represented by strictly positive continuous density functions $\mu, \nu : \mathcal{M} \to \mathbb{R}_{>0}$, where $\mu$ by convention represents the target (unknown) distribution and $\nu$ represents the source (prior) distribution. $\mu, \nu$ are probability densities in the sense their integral w.r.t. the Riemannian volume form is one, i.e., $\int_{\mathcal{M}} \mu dV = 1 = \int_{\mathcal{M}} \nu dV$. It is convenient to consider the volume forms that correspond to the probability measures, namely $\hat{\mu} = \mu dV$ and $\hat{\nu} = \nu dV$. Volume forms are differential $n$-forms that can be integrated over subdomains of $\mathcal{M}$, for example, $p_\nu(\mathcal{A}) = \int_{\mathcal{A}} \hat{\nu}$ is the probability of the event $\mathcal{A} \subset \mathcal{M}$.

**Continuous Normalizing Flows (CNF) on manifolds** operate by transforming a simple source distribution through a map $\Phi$ into a highly complex and multimodal target distribution. A manifold CNF, $\Phi : \mathcal{M} \to \mathcal{M}$, is an orientation preserving diffeomorphism from the manifold to itself (Mathieu and Nickel, 2020; Lou et al., 2020; Falorsi and Forré, 2020). A smooth map $\Phi : \mathcal{M} \to \mathcal{M}$ can be used to *pull-back* the target $\hat{\mu}$ according to the formula:

$$(\Phi^*\hat{\mu})_z(v_1, \ldots, v_n) = \hat{\mu}_{\Phi(z)}(D\Phi_z(v_1), \ldots, D\Phi_z(v_n)), \tag{1}$$

where $v_1, \ldots, v_n \in T_z\mathcal{M}$ are arbitrary tangent vectors, $D\Phi_z : T_z\mathcal{M} \to T_{\Phi(z)}\mathcal{M}$ is the differential of $\Phi$, namely a linear map between the tangent spaces to $\mathcal{M}$ at the points $z$ and $\Phi(z)$, respectively. By pulling-back $\hat{\mu}$ according to $\Phi$ and asking it to equal to the prior density $\nu$, we get the manifold version of the standard *normalizing equation*:

$$\hat{\nu} = \Phi^*\hat{\mu}. \tag{2}$$

If the normalizing equation holds, then for an event $\mathcal{A} \subset \mathcal{M}$ we have that

$$p_\nu(\mathcal{A}) = \int_{\mathcal{A}} \hat{\nu} = \int_{\mathcal{A}} \Phi^* \hat{\mu} = \int_{\Phi(\mathcal{A})} \hat{\mu} = p_\mu(\Phi(\mathcal{A})).$$

Therefore, given a random variable $z$ distributed according to $\nu$, then $x = \Phi(z)$ is distributed according to $\mu$, and $\Phi$ is the *generator*.

One way to construct a CNF $\Phi$ is by solving an ordinary differential equation (ODE) (Chen et al., 2018; Mathieu and Nickel, 2020). Given a time-dependent vector field $v_t \in \mathfrak{X}(\mathcal{M})$ with $t \in [0, 1]$, a one-parameter family of diffeomorphisms (CNFs) $\Phi_t : [0, 1] \times \mathcal{M} \to \mathcal{M}$ is defined by

$$\frac{d}{dt}\Phi_t = v_t(\Phi_t), \tag{3}$$

where this ODE is initialized with the identity transformation, i.e., for all $x \in \mathcal{M}$ we initialize $\Phi_0(x) = x$. The CNF is then defined by $\Phi(x) = \Phi_1(x)$.

**Example: Euclidean CNF.** Let us show how the above notions boil down to standard Euclidean CNF for the choice of $\mathcal{M} = \mathbb{R}^n$, and the standard Euclidean metric; we denote $z = (z^1, \ldots, z^n) \in \mathbb{R}^n$. The Riemannian volume form in this case is $dz = dz^1 \wedge dz^2 \wedge \cdots \wedge dz^n$. Furthermore, $\hat{\mu}(z) = \mu(z)dz$ and $\hat{\nu}(z) = \nu(z)dz$. The pull-back formula (equation 1) in coordinates (see e.g., Proposition 14.20 in Lee (2013)) is

$$\Phi^* \hat{\mu}(z) = \mu(\Phi(z)) \det(D\Phi_z) dz,$$

where $D\Phi_z$ is the matrix of partials of $\Phi$ at point $z$, $(D\Phi_z)_{ij} = \frac{\partial \Phi^i}{\partial z^j}(z)$. Plugging this in equation 2 we get the Euclidean normalizing equation:

$$\nu(z) = \mu(\Phi(z)) \det(D\Phi_z). \tag{4}$$

## 3 Moser Flow

Moser (1965) and Dacorogna and Moser (1990) suggested a method for solving the normalizing equation, that is equation 2. Their method explicitly constructs a vector field $v_t$, and the flow it defines via equation 3 is guaranteed to solve equation 2. We start by introducing the method, adapted to our needs, followed by its application to generative modeling. We will use notations introduced above.

### 3.1 Solving the normalizing equation

Moser's approach to solving equation 2 starts by interpolating the source and target distributions. That is, choosing an interpolant $\alpha_t : [0, 1] \times \mathcal{M} \to \mathbb{R}_{>0}$, such that $\alpha_0 = \nu$, $\alpha_1 = \mu$, and $\int_{\mathcal{M}} \alpha_t dV = 1$ for all $t \in [0, 1]$. Then, a time-dependent vector field $v_t \in \mathfrak{X}(\mathcal{M})$ is defined so that for each time $t \in [0, 1]$, the flow $\Phi_t$ defined by equation 3 satisfies the *continuous normalization equation*:

$$\Phi_t^* \hat{\alpha}_t = \hat{\alpha}_0, \tag{5}$$

where $\hat{\alpha}_t = \alpha_t dV$ is the volume form corresponding to the density $\alpha_t$. Clearly, plugging $t = 1$ in the above equation provides a solution to equation 2 with $\Phi = \Phi_1$. As it turns out, considering the continuous normalization equation simplifies matters and the sought after vector field $v_t$ is constructed as follows. First, solve the partial differential equation (PDE) over the manifold $\mathcal{M}$

$$\operatorname{div}(u_t) = -\frac{d}{dt}\alpha_t, \tag{6}$$

where $u_t \in \mathfrak{X}(\mathcal{M})$ is an unknown time-dependent vector field, and div is the Riemannian generalization to the Euclidean divergence operator, $\operatorname{div}_E = \nabla \cdot$. This manifold divergence operator is defined by replacing the directional derivative of the Euclidean space with its Riemannian version, namely the covariant derivative,

$$\operatorname{div}(u) = \sum_{i=1}^{n} \langle \nabla_{e_i} u, e_i \rangle_g, \tag{7}$$

where $\{e_i\}_{i=1}^n$ is an orthonormal frame according to the Riemannian metric $g$, and $\nabla_\xi u$ is the Riemannian covariant derivative. Note that here we assume that $\mathcal{M}$ is boundaryless, otherwise we need $u_t$ to be also tangent to the boundary of $\mathcal{M}$. Second, define

$$v_t = \frac{u_t}{\alpha_t}. \tag{8}$$

Theorem 2 in Moser (1965) implies:

**Theorem 1** (Moser). *The diffeomorphism $\Phi = \Phi_1$, defined by the ODE in equation 3 and vector field $v_t$ in equation 8 solves the normalization equation, i.e., equation 2.*

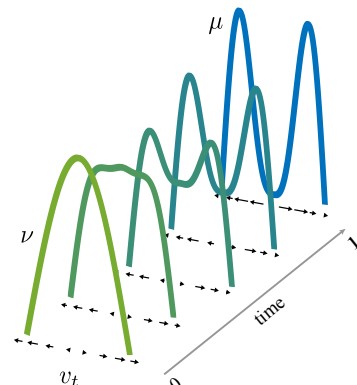

The proof of this theorem in our case is provided in the supplementary for completeness. A simple choice for the interpolant $\alpha_t$ that we use in this paper was suggested in Dacorogna and Moser (1990):

$$\alpha_t = (1-t)\nu + t\mu. \tag{9}$$

The time derivative of this interpolant, i.e., $\frac{d}{dt}\alpha_t = \mu - \nu$, does not depend on $t$. Therefore the vector field can be chosen to be constant over time, $u_t = u$, and the PDE in equation 6 takes the form

$$\mathrm{div}(u) = \nu - \mu, \tag{10}$$

and consequently $v_t$ takes the form

$$v_t = \frac{u}{(1-t)\nu + t\mu}. \tag{11}$$

Figure 1 shows a one dimensional illustration of Moser Flow.

Figure 1: 1D example of Moser Flow: source density $\nu$ in green, target $\mu$ in blue. The vector field $v_t$ (black) is guaranteed to push $\nu$ to interpolated density $\alpha_t$ at time $t$, i.e., $(1-t)\nu+t\mu$.

### 3.2 Generative model utilizing Moser Flow

We next utilize MF to define our generative model. Our model (learned) density $\bar{\mu}$ is motivated from equation 10 and is defined by

$$\bar{\mu} = \nu - \mathrm{div}(u), \tag{12}$$

where $u \in \mathfrak{X}(\mathcal{M})$ is the degree of freedom of the model. We model this degree of freedom, $u$, with a deep neural network, more specifically, Multi-Layer Perceptron (MLP). We denote $\theta \in \mathbb{R}^p$ the learnable parameters of $u$. We start by noting that, by construction, $\bar{\mu}$ has a unit integral over $\mathcal{M}$:

**Lemma 1.** *If $\mathcal{M}$ has no boundary, or $\mathbf{u}|_{\partial\mathcal{M}} \in \mathfrak{X}(\partial\mathcal{M})$, then $\int_{\mathcal{M}} \bar{\mu}dV = 1$.*

This lemma is proved in the supplementary and a direct consequence of Stokes' Theorem. If $\bar{\mu} > 0$ over $\mathcal{M}$ then, together with the fact that $\int_{\mathcal{M}} \bar{\mu}dV = 1$ (Lemma 1), it is a probability density over $\mathcal{M}$. Consequently, Theorem 1 implies that $\bar{\mu}$ is realized by a CNF defined via $v_t$:

**Corollary 1.** *If $\bar{\mu} > 0$ over $\mathcal{M}$ then $\bar{\mu}$ is a probability distribution over $\mathcal{M}$, and is generated by the flow $\Phi = \Phi_1$, where $\Phi_t$ is the solution to the ODE in equation 3 with the vector field $v_t \in \mathfrak{X}(\mathcal{M})$ defined in equation 11.*

Since $\bar{\mu} > 0$ is an open constraint and is not directly implementable, we replace it with the closed constraint $\bar{\mu} \geq \epsilon$, where $\epsilon > 0$ is a small hyper-parameter. We define

$$\bar{\mu}_+(x) = \max\{\epsilon, \bar{\mu}(x)\}; \quad \bar{\mu}_-(x) = \epsilon - \min\{\epsilon, \bar{\mu}(x)\}.$$

As can be readily verified:

$$\bar{\mu}_+, \bar{\mu}_- \geq 0, \ \text{and} \ \bar{\mu} = \bar{\mu}_+ - \bar{\mu}_-. \tag{13}$$

We are ready to formulate the loss for training the generative model. Consider an unknown target distribution $\mu$, provided to us as a set of i.i.d. observations $\mathcal{X} = \{x_i\}_{i=1}^m \subset \mathcal{M}$. Our goal is to maximize the likelihood of the data $\mathcal{X}$ while making sure $\bar{\mu} \geq \epsilon$. We therefore consider the following loss:

$$\ell(\theta) = -\mathbb{E}_\mu \log \bar{\mu}_+(x) + \lambda \int_{\mathcal{M}} \bar{\mu}_-(x)dV \tag{14}$$

where $\lambda$ is a hyper-parameter. The first term in the loss is approximated by the empirical mean computed with the observations $\mathcal{X}$, i.e.,

$$\mathbb{E}_{x \sim \mu} \log \bar{\mu}_+(x) \approx \frac{1}{m} \sum_{i=1}^{m} \log \bar{\mu}_+(x_i).$$

This term is merely the negative log likelihood of the observations.

The second term in the loss penalizes the deviation of $\bar{\mu}$ from satisfying $\bar{\mu} \geq \epsilon$. According to Theorem 1, this measures the deviation of $\bar{\mu}$ from being a density function and realizing a CNF. One point that needs to be verified is that the combination of these two terms does not push the minimum away from the target density $\mu$. This can be verified with the help of the generalized Kullback–Leibler (KL) divergence providing a distance measure between arbitrary positive functions $f, g : \mathcal{M} \to \mathbb{R}_{>0}$:

$$D(f,g) = \int_{\mathcal{M}} f \log\left(\frac{f}{g}\right) dV - \int_{\mathcal{M}} f dV + \int_{\mathcal{M}} g dV. \tag{15}$$

Using the generalized KL, we can now compute the distance between the positive part of our model density, i.e., $\bar{\mu}_+$, and the target density:

$$D(\mu, \bar{\mu}_+) = \mathbb{E}_\mu \log\left(\frac{\mu}{\bar{\mu}_+}\right) - \int_{\mathcal{M}} \mu dV + \int_{\mathcal{M}} \bar{\mu}_+ dV$$

$$= \mathbb{E}_\mu \log \mu - \mathbb{E}_\mu \log \bar{\mu}_+ + \int_{\mathcal{M}} \bar{\mu}_- dV$$

where in the second equality we used Lemma 1. The term $\mathbb{E}_\mu \log \mu$ is the negative entropy of the unknown target distribution $\mu$. The loss in equation 14 equals $D(\mu, \bar{\mu}_+) - \mathbb{E}_\mu \log \mu + (\lambda - 1) \int_{\mathcal{M}} \bar{\mu}_- dV$. Therefore, if $\lambda \geq 1$, and $\min_{x \in \mathcal{M}} \mu(x) > \epsilon$ (we use the compactness of $\mathcal{M}$ to infer existence of such a minimal positive value), then the unique minimum of the loss in equation 14 is the target density, i.e., $\bar{\mu} = \mu$. Indeed, the minimal value of this loss is $-\mathbb{E}_\mu \log \mu$ and it is achieved by setting $\bar{\mu} = \mu$. Uniqueness follows by considering an arbitrary minimizer $\bar{\mu}$. Since such a minimizer satisfies $D(\mu, \bar{\mu}_+) = 0$ and $\int_{\mathcal{M}} \bar{\mu}_- dV = 0$, necessarily $\bar{\mu} = \mu$. We proved:

**Theorem 2.** *For $\lambda \geq 1$ and sufficiently small $\epsilon > 0$, the unique minimizer of the loss in equation 14 is $\bar{\mu} = \mu$.*

**Variation of the loss.** Lemma 1 and equation 13 imply that $\int_{\mathcal{M}} \bar{\mu}_+ dV = \int_{\mathcal{M}} \bar{\mu}_- dV + 1$. Therefore, an equivalent loss to the one presented in equation 14 is:

$$\ell(\theta) = -\mathbb{E}_\mu \log \bar{\mu}_+(x) + \lambda_- \int_{\mathcal{M}} \bar{\mu}_- dV + \lambda_+ \int_{\mathcal{M}} \bar{\mu}_+ dV \tag{16}$$

with $\lambda_- + \lambda_+ \geq 1$. Empirically we found this loss favorable in some cases (i.e., with $\lambda_+ > 0$).

**Integral approximation.** The integral $\int_{\mathcal{M}} \bar{\mu}_- dV$ in the losses in equation 16 and 14 is approximated by considering a set $\mathcal{Y} = \{y_j\}_{j=1}^{l}$ of i.i.d. samples according to some distribution $\eta$ over $\mathcal{M}$ and taking a Monte Carlo estimate

$$\int_{\mathcal{M}} \bar{\mu}_- dV \approx \frac{1}{l} \sum_{j=1}^{l} \frac{\bar{\mu}_-(y_j)}{\eta(y_j)}. \tag{17}$$

$\int_{\mathcal{M}} \bar{\mu}_+ dV$ is approximated similarly. In this paper we opted for the simple choice of taking $\eta$ to be the uniform distribution over $\mathcal{M}$.

## 4  Generative modeling over Euclidean submanifolds

In this section, we adapt the Moser Flow (MF) generative model to submanifolds of Euclidean spaces. That is we consider an orientable, compact, boundaryless, connected $n$-dimensional submanifold $\mathcal{M} \subset \mathbb{R}^d$, where $n < d$. Examples include implicit surfaces and manifolds (i.e., preimage of a regular value of a smooth function), as well as triangulated surfaces and manifold simplicial complexes. We

denote points in $\mathbb{R}^d$ (and therefore in $\mathcal{M}$) with $\boldsymbol{x}, \boldsymbol{y} \in \mathbb{R}^d$. As the Riemannian metric of $\mathcal{M}$ we take the induced metric from $\mathbb{R}^d$; that is given arbitrary tangent vectors $\boldsymbol{v}, \boldsymbol{u} \in T_{\boldsymbol{x}}\mathcal{M}$ the metric is defined by $\langle \boldsymbol{v}, \boldsymbol{u} \rangle_g = \langle \boldsymbol{v}, \boldsymbol{u} \rangle$, where the latter is the Euclidean inner product. We denote by $\pi : \mathbb{R}^d \to \mathcal{M}$ the closest point projection on $\mathcal{M}$, i.e., $\pi(\boldsymbol{x}) = \min_{\boldsymbol{y} \in \mathcal{M}} \|\boldsymbol{x} - \boldsymbol{y}\|$, with $\|\boldsymbol{y}\|^2 = \langle \boldsymbol{y}, \boldsymbol{y} \rangle$ the Euclidean norm in $\mathbb{R}^d$. Lastly, we denote by $\boldsymbol{P}_{\boldsymbol{x}} \in \mathbb{R}^{d \times d}$ the orthogonal projection matrix on the tangent space $T_{\boldsymbol{x}}\mathcal{M}$; in practice if we denote by $\boldsymbol{N} \in \mathbb{R}^{d \times k}$ the matrix with orthonormal columns spanning $N_{\boldsymbol{x}}\mathcal{M} = (T_x\mathcal{M})^\perp$ (i.e., the normal space to $\mathcal{M}$ at $\boldsymbol{x}$) then, $\boldsymbol{P}_{\boldsymbol{x}} = \boldsymbol{I} - \boldsymbol{N}\boldsymbol{N}^T$.

We parametrize the vector field $\boldsymbol{u}$ required for our MF model (in equation 12) by defining a vector field $\boldsymbol{u} \in \mathfrak{X}(\mathbb{R}^d)$ such that $\boldsymbol{u}|_{\mathcal{M}} \in \mathfrak{X}(\mathcal{M})$. We define

$$\boldsymbol{u}(\boldsymbol{x}) = \boldsymbol{P}_{\pi(\boldsymbol{x})}\boldsymbol{v}_\theta(\pi(\boldsymbol{x})), \tag{18}$$

where $\boldsymbol{v}_\theta : \mathbb{R}^d \to \mathbb{R}^d$ is an MLP with Softplus activation ($\beta = 100$) and learnable parameters $\theta \in \mathbb{R}^p$. By construction, for $\boldsymbol{x} \in \mathcal{M}$, $\boldsymbol{u}(\boldsymbol{x}) \in T_{\boldsymbol{x}}\mathcal{M}$.

To realize the generative model, we need to compute the divergence $\mathrm{div}(\boldsymbol{u}(\boldsymbol{x}))$ for $\boldsymbol{x} \in \mathcal{M}$ with respect to the Riemannian manifold $\mathcal{M}$ and metric $g$. The vector field $\boldsymbol{u}$ in equation 18 is constant along normal directions to the manifold at $\boldsymbol{x}$ (since $\pi(\boldsymbol{x})$ is constant in normal directions). If $\boldsymbol{n} \in N_{\boldsymbol{x}}\mathcal{M}$, then in particular

$$\frac{d}{dt}\bigg|_{t=0} \boldsymbol{u}(\boldsymbol{x} + t\boldsymbol{n}) = 0. \tag{19}$$

We call vector fields that satisfy equation 19 infinitesimally constant in the normal direction. As we show next, such vector fields $\boldsymbol{u} \in \mathfrak{X}(\mathcal{M})$ have the useful property that their divergence along the manifold $\mathcal{M}$ coincides with their Euclidean divergence in the ambient space $\mathbb{R}^d$:

**Lemma 2.** *If $\boldsymbol{u} \in \mathfrak{X}(\mathbb{R}^d)$, $\boldsymbol{u}|_{\mathcal{M}} \in \mathfrak{X}(\mathcal{M})$ is infinitesimally constant in normal directions of $\mathcal{M}$, then for $\boldsymbol{x} \in \mathcal{M}$, $\mathrm{div}(\boldsymbol{u}(\boldsymbol{x})) = \mathrm{div}_E(\boldsymbol{u}(\boldsymbol{x}))$, where $\mathrm{div}_E$ denotes the standard Euclidean divergence.*

This lemma simply means we can compute the Euclidean divergence of $\boldsymbol{u}$ in our implementation. Given a set of observed data $\mathcal{X} = \{\boldsymbol{x}_i\}_{i=1}^m \subset \mathcal{M} \subset \mathbb{R}^d$, and a set of uniform i.i.d. samples $\mathcal{Y} = \{\boldsymbol{y}_j\}_{j=1}^l \subset \mathcal{M}$ over $\mathcal{M}$, our loss (equation 14) takes the form

$$\ell(\theta) = -\frac{1}{m}\sum_{i=1}^m \log\max\{\epsilon, \nu(\boldsymbol{x}_i) - \mathrm{div}_E\boldsymbol{u}(\boldsymbol{x}_i)\} + \frac{\lambda'_-}{l}\sum_{j=1}^l \Big(\epsilon - \min\{\epsilon, \nu(\boldsymbol{y}_j) - \mathrm{div}_E\boldsymbol{u}(\boldsymbol{y}_j)\}\Big),$$

where $\lambda'_- = \lambda_- V(M)$. We note the volume constant can be ignored by considering an un-normalized source density $V(M)\nu \equiv 1$, see supplementary for details. The loss in equation 16 is implemented similarly, namely, we add the empirical approximation of $\lambda_+ \int_{\mathcal{M}} \bar{\mu}_+ dV$.

We conclude this section by stating that the MF generative model over Euclidean submanifolds (defined with equations 12 and 18) is universal. That is, MFs can generate, arbitrarily well, any continuous target density $\mu$ on a submanifold manifold $\mathcal{M} \subset \mathbb{R}^d$.

**Theorem 3.** *Given an orientable, compact, boundaryless, connected, differentiable $n$-dimensional submanifold $\mathcal{M} \subset \mathbb{R}^d$, $n < d$, and a target continuous probability density $\mu : \mathcal{M} \to \mathbb{R}_{>0}$, there exists for each $\epsilon > 0$ an MLP $\boldsymbol{v}_\theta$ and a choice of weights $\theta$ so that $\bar{\mu}$ defined by equations 12 and 18 satisfies*

$$\max_{\boldsymbol{x} \in \mathcal{M}} |\mu(\boldsymbol{x}) - \bar{\mu}(\boldsymbol{x})| < \epsilon.$$

## 5  Experiments

In all experiments, we modeled a manifold vector field as a multi-layer perceptron (MLP) $\boldsymbol{u}_\theta \in \mathfrak{X}(\mathcal{M})$, with parameters $\theta$. All models were trained using Adam optimizer (Kingma and Ba, 2014), and in all neural networks the activation is Softplus with $\beta = 100$. Unless stated otherwise, we set $\lambda_+ = 0$. We used an exact calculation of the divergence $\mathrm{div}_E(\boldsymbol{u}(\boldsymbol{x}))$. We experimented with two kinds of manifolds.

**Flat Torus.** To test our method on Euclidean 2D data, we used $\mathcal{M}$ as the flat torus, that is the unit square $[-1, 1]^2$ with opposite edges identified. This defines a manifold with no boundary which is locally isometric to the Euclidean plane. Due to this local isometry the Riemannian divergence on the flat

| input data | samples | density | input data | samples | density |
|---|---|---|---|---|---|

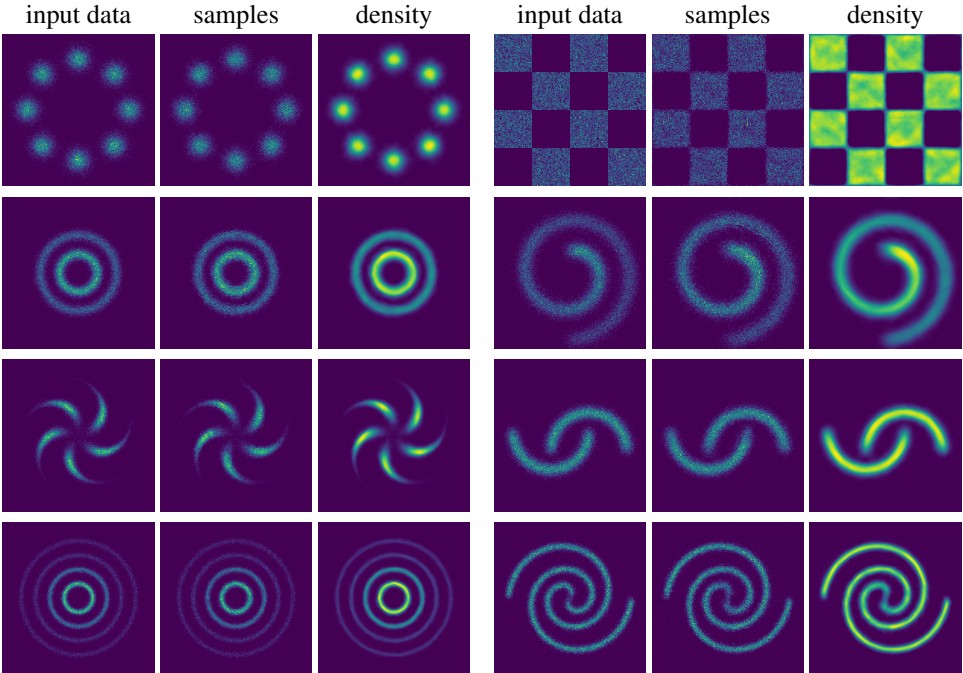

Figure 2: Moser Flow trained on 2D datasets. We show generated samples and learned density $\bar{\mu}_+$.

torus is equivalent to the Euclidean divergence, $\mathrm{div} = \mathrm{div}_E$. To make $\boldsymbol{u}_\theta$ a well defined smooth vector field in $\mathcal{M}$ we use periodic positional encoding, namely $\boldsymbol{u}_\theta(\boldsymbol{x}) = \boldsymbol{v}_\theta(\tau(\boldsymbol{x}))$, where $\boldsymbol{v}_\theta$ is a standard MLP and $\tau : \mathbb{R}^2 \to \mathbb{R}^{4k}$ is defined as $\tau(\boldsymbol{x}) = (\cos(\omega_1 \pi \boldsymbol{x}), \sin(\omega_1 \pi \boldsymbol{x}), ..., \cos(\omega_k \pi \boldsymbol{x}), \sin(\omega_k \pi \boldsymbol{x}))$, where $w_i = i$, and $k$ is a hyper-parameter that is application dependent. Since any periodic function can be approximated by a polynomial acting on $e^{i\pi \boldsymbol{x}}$, even for $k = 1$ this is a universal model for continuous functions on the torus. As described by Tancik et al. (2020), adding extra features can help with learning higher frequencies in the data. To solve an ODE on the torus we simply solve it for the periodic function and wrap the result back to $[-1, 1]^2$.

**Implicit surfaces.** We experiment with surfaces as submanifolds of $\mathbb{R}^3$. We represent a surface as the zero level set of a Signed Distance Function (SDF) $f : \mathbb{R}^3 \to \mathbb{R}$. We experimented with two surfaces. First, the sphere, represented with the SDF $f(\boldsymbol{x}) = \|\boldsymbol{x}\| - 1$, and second, the Stanford Bunny surface, representing a general curved surface and represented with an SDF learned with (Gropp et al., 2020) from point cloud data. To model vector fields on an implicit surface we follow the general equation 18, where for SDFs

$$\pi(\boldsymbol{x}) = \boldsymbol{x} - f(\boldsymbol{x})\nabla f(\boldsymbol{x}), \quad \text{and} \quad \boldsymbol{P_x} = \boldsymbol{I} - \nabla f(\boldsymbol{x})\nabla f(\boldsymbol{x})^T.$$

In the supplementary, we detail how to replace the global projection $\pi(\boldsymbol{x})$ with a local one, for cases the SDF is not exact.

### 5.1 Toy distributions

First, we consider a collection of challenging toy 2D datasets explored in prior works (Chen et al., 2020; Huang et al., 2021). We scale samples to lie in the flat torus $[-1, 1]^2$ and use $k = 1$ for the positional encoding. Figure 2 depicts the input data samples, the generated samples *after* training, and the learned distribution $\bar{\mu}$. In the top six datasets, the MLP architecture used for Moser Flows consists of 3 hidden layers with 256 units each, whereas in the bottom two we used 4 hidden layers with 256 neurons due to the higher complexity of these distributions. We set $\lambda_- = 2$.

### 5.2 Choice of hyper-parameter $\lambda$

We test the effect of the hyper-parameter $\lambda \geq 1$ on the learned density. Figure 3 shows, for different values of $\lambda$, our density estimation $\mu_+$, the push-forward density $\Phi_* \nu$, and their absolute difference. To evaluate $\Phi_* \nu$ from the vector field $v_t$, we solve an ODE as advised in Grathwohl et al. (2018).

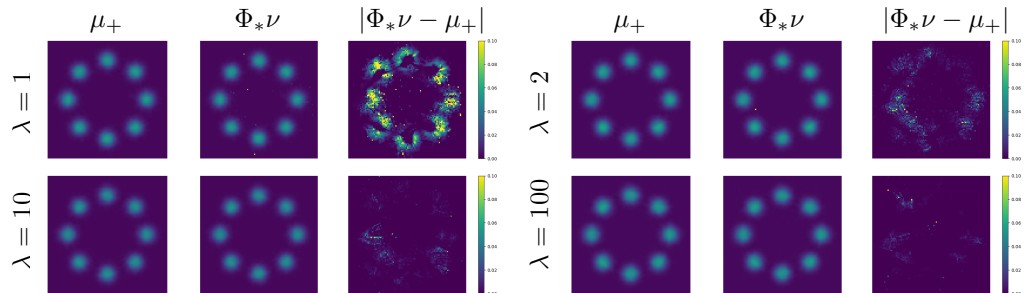

Figure 3: As $\lambda$ is increased, the closer $\bar{\mu}_+$ is to the generated density $\Phi_*\nu$; column titled $|\bar{\mu}_+ - \Phi_*\nu|$ shows the absolute pointwise difference between the two; note that some of the errors in the $|\bar{\mu}_+ - \Phi_*\nu|$ column are due to numerical inaccuracies in the ODE solver used to calculate $\Phi_*\nu$.

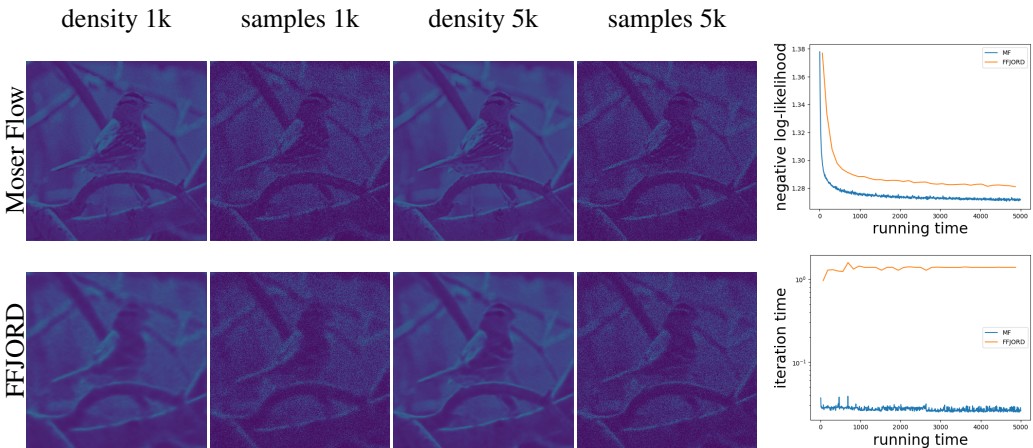

Figure 4: Comparing learned density and generated samples with MF and FFJORD at different times (in k-sec); top right shows NLL scores for both MF and FFJORD at different times; bottom right shows time per iteration (in $\log$-scale, sec) as a function of total running time (in sec); FFJORD iterations take longer as training progresses; A second example of the same experiment on a different image is provided in the supplementary. Flickr image (license CC BY 2.0): Bird by Flickr user "lakeworth" `https://www.flickr.com/photos/lakeworth/46657879995/`.

As expected, higher values of $\lambda$ lead to closer modeled density $\bar{\mu}_+$ and $\Phi_*\nu$. This is due to the fact that a higher value of $\lambda$ leads to a lower value of $\int_{\mathcal{M}} \bar{\mu}_-$, meaning $\bar{\mu}$ is a better representation of a probability density. Nonetheless, even for $\lambda = 1$ the learned and generated density are rather consistent.

## 5.3 Time evaluations

To compare our method to Euclidean CNF methods, we compare Moser Flow with FFJORD (Grathwohl et al., 2018) on the flat torus. We consider a challenging density with high frequencies obtained via 800x600 images (Figure 4). We generate a new batch every iteration by sampling each pixel location with probability which is proportional to the pixel intensity. The architectures of both $v_\theta$ and the vector field defined in FFJORD are the same, namely an MLP with 4 hidden layers of 256 neurons each. To capture the higher frequencies in the image we use a positional encoding with $k = 8$ for both methods. We used a batch size of 10k. We used learning rate of 1e-5 for Moser Flow and 1e-4 for FFJORD. We used $\lambda_- = 2$. Learning was stopped after 5k seconds. Figure 4 presents the results. Note that Moser Flow captures high-frequency details better than FFJORD. This is also expressed in the graph on the top right, showing how the NLL decreases faster for MF than FFJORD. Furthermore, as can be inspected in the per iteration time graph on the bottom-right, MF per iteration time does not increase during training, and is roughly 1-2 order of magnitudes faster than FFJORD iteration.

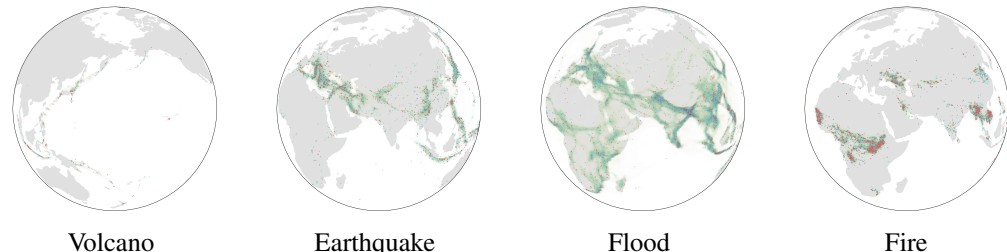

| | Volcano | Earthquake | Flood | Fire |
|---|---|---|---|---|

Figure 5: Moser Flow trained on earth sciences data gathered by Mathieu and Nickel (2020). The learned density is colored green-blue (blue indicates larger values); Blue and red dots represent training and testing datapoints, respectively. See Table 1 for matching quantitative results.

| | Volcano | Earthquake | Flood | Fire |
|---|---|---|---|---|
| Mixture vMF | $-0.31_{\pm 0.07}$ | $0.59_{\pm 0.01}$ | $1.09_{\pm 0.01}$ | $-0.23_{\pm 0.02}$ |
| Stereographic | $-0.64_{\pm 0.20}$ | $0.43_{\pm 0.04}$ | $0.99_{\pm 0.04}$ | $-0.40_{\pm 0.06}$ |
| Riemannian | $-0.97_{\pm 0.15}$ | $0.19_{\pm 0.04}$ | $0.90_{\pm 0.03}$ | $-0.66_{\pm 0.05}$ |
| Moser Flow (MF) | $\mathbf{-2.02}_{\pm 0.42}$ | $\mathbf{-0.09}_{\pm 0.02}$ | $\mathbf{0.62}_{\pm 0.04}$ | $\mathbf{-1.03}_{\pm 0.03}$ |
| Data size | 829 | 6124 | 4877 | 12810 |

Table 1: Negative log-likelihood scores of the earth sciences datasets.

## 5.4 Earth and climate science data

We evaluate our model on the earth and climate datasets gathered in Mathieu and Nickel (2020). The projection operator $\pi$ in this case is simply $\pi(\boldsymbol{x}) = \frac{\boldsymbol{x}}{\|\boldsymbol{x}\|}$. We parameterize $\boldsymbol{v}_\theta$ as an MLP with 6 hidden layers of 512 neurons each. We used full batches for the NLL loss and batches of size 150k for our integral approximation. We trained for 30k epochs, with learning rate of 1e-4. We used $\lambda_- = 100$. The quantitative NLL results are reported in Table 1 and qualitative visualizations in 5. Note that we produce NLL scores smaller than the runner-up method by a large margin.

## 5.5 Curved surfaces

We trained an SDF $f$ for the Stanford Bunny surface $\mathcal{M}$ using the method in Gropp et al. (2020). To generate uniform ($\nu$) and data ($\mu$) samples over $\mathcal{M}$ we first extract a mesh $\mathcal{M}'$ from $f$ using the Marching Cubes algorithm (Lorensen and Cline, 1987) setting its resolution to 100x100x100. Then, to randomly choose a point uniformly from $\mathcal{M}'$ we first randomly choose a face of the mesh with probability proportional to its area, and then randomly choose a point uniformly within that face. For target $\mu$ we used clamped manifold harmonics to create a sequence of densities with increased complexity. To that end, we first computed the $k$-th eigenfunction of the Laplace-Beltrami operator over $\mathcal{M}'$ (we provide details on this computation in the supplementary), for the frequencies (eigenvalues) $k \in \{10, 50, 500\}$. Next, we sampled the eigenfunctions at the faces' centers, clamped their negative values, and normalized to get discrete probability densities over the faces of $\mathcal{M}'$. Then, to sample a point, we first choose a face at random based on this probability, and then random a point uniformly within that face. We take 500k i.i.d. samples of this distribution as our dataset. We take $\boldsymbol{v}_\theta$ to be an MLP with 6 hidden layers of dimension 512. We use batch size of 10k for both the NLL loss and for the integral approximation; we ran for 1000 epochs with learning rate of 1e-4. We used $\lambda_- = \lambda_+ = 1$. Figure 6 depict the results. Note that Moser Flow is able to learn the surface densities for all three frequencies.

## 6 Related Work

In the following, we discuss related work on normalizing flows for manifold-valued data. On a high level, such methods can be divided into *Parametric* versus *Riemannian* methods. Parametric methods consist of a normalizing flow in the Euclidean space $\mathbb{R}^n$, pushed-forward onto the manifold through an invertible map $\psi : \mathbb{R}^n \to \mathcal{M}$. However, to globally represent the manifold, $\psi$ needs to be a homeomorphism implying that $\mathcal{M}$ and $\mathbb{R}^n$ are topologically equivalent, limiting the scope of that

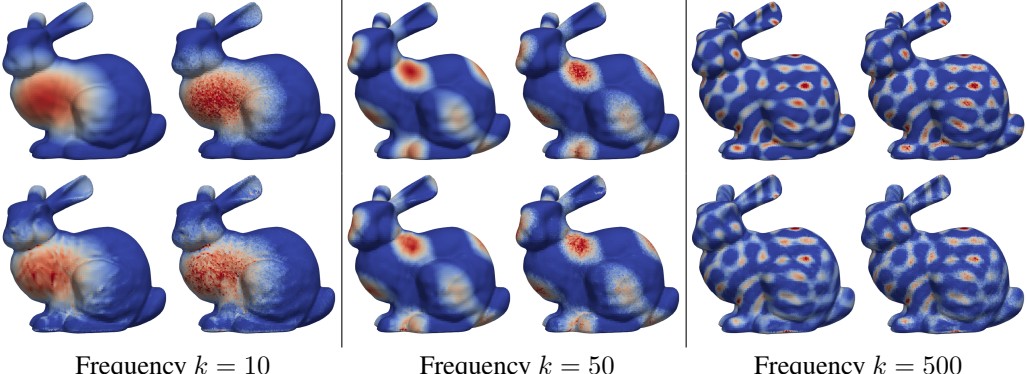

Frequency $k = 10$      Frequency $k = 50$      Frequency $k = 500$

Figure 6: Moser Flow trained on a curved surface (Stanford Bunny). We show three different target distribution with increasing frequencies, where for each frequency we depict (clockwise from top-left): target density, data samples, generated samples, and learned density.

approach. Existing methods in this class are often based on the exponential map $\exp_x : T_x\mathcal{M} \cong \mathbb{R}^n \to \mathcal{M}$ of a manifold. This leads to the so called *wrapped* distributions. This approach has been taken, for instance, by Falorsi et al. (2019) and Bose et al. (2020) to parametrize probability distributions on Lie groups and hyperbolic space. However, Parametric methods based on the exponential map often lead to numerical and computational challenges. For instance, in compact manifolds (e.g., spheres or the SO(3) group) computing the density of *wrapped* distributions requires an infinite summation. On the hyperboloid, on the other hand, the exponential map is numerically not well-behaved far away from the origin (Dooley and Wildberger, 1993; Al-Mohy and Higham, 2010).

In contrast to Parametric methods, Riemannian methods operate directly on the manifold itself and, as such, avoid numerical instabilities that arise from the mapping onto the manifold. Early work in this class of models proposed transformations along geodesics on the hypersphere by evaluating the exponential map at the gradient of a scalar manifold function (Sei, 2011). Rezende et al. (2020) introduced *discrete Riemannian* flows for hyperspheres and torii based on Möbius transformations and spherical splines. Mathieu and Nickel (2020) introduced *continuous* flows on general Riemannian manifolds (RCNF). In contrast to discrete flows (e.g., Bose et al., 2020; Rezende et al., 2020), such time-continuous flows alleviate the previous topological constraints by parametrizing the flow as the solution to an ODE over the manifold (Grathwohl et al., 2018). Concurrently to RCNF, Lou et al. (2020) and Falorsi and Forré (2020) proposed related extensions of neural ODEs to smooth manifolds. Moser Flow also generates a CNF, however by limiting the flow space (albeit, not the generated distributions) it allows expressing the learned distribution as the divergence of a vector field.

## 7 Discussion and limitations

We introduced Moser Flow, a generative model in the family of CNFs that represents the target density using the divergence operator applied to a vector valued neural network. The main benefits of MF stems from the simplicity and locality of the divergence operator. MFs circumvent the need to solve an ODE in the training process, and are thus applicable on a broad class of manifolds. Theoretically, we prove MF is a universal generative model, able to (approximately) generate arbitrary positive target densities from arbitrary positive prior densities. Empirically, we show MF enjoys favorable computational speed in comparison to previous CNF models, improves density estimation on spherical data compared to previous work by a large margin, and for the first time facilitate training a CNF over a general curved surface.

One important future work direction, and a current limitation, is scaling of MF to higher dimensions. This challenge can be roughly broken to three parts: First, the model probabilities $\bar{\mu}$ should be computed/approximated in log-scale, as probabilities are expected to decrease exponentially with the dimension. Second, the variance of the numerical approximations of the integral $\int_{\mathcal{M}} \bar{\mu}_- dV$ will increase significantly in high dimensions and needs to be controlled. Third, the divergence term, $\mathrm{div}(u)$, is too costly to be computed exactly in high dimensions and needs to be approximated, similarly to other CNF approaches. Finally, our work suggests a novel generative model, and similarly to other generative models can be potentially used for generation of fake data and amplify harmful biases in the dataset. Mitigating such harms is an active and important area of ongoing research.

## Acknowledgments

NR is supported by the European Research Council (ERC Consolidator Grant, "LiftMatch" 771136), the Israel Science Foundation (Grant No. 1830/17), and Carolito Stiftung (WAIC).

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
