# Moser Flow: Divergence-based Generative Modeling on Manifolds
## Supplementary

**Noam Rozen**[1]     **Aditya Grover**[2,3]     **Maximilian Nickel**[2]     **Yaron Lipman**[1,2]

[1]Weizmann Institute of Science     [2]Facebook AI Research     [3]UCLA

## A   Proof of Moser's Theorem.

We will review here the proof of Moser Theorem 1; for more details see Moser's original paper (Moser, 1965) or Lang (2012), Chapter 18 section 2. Let $\hat{\alpha}_t = \alpha_t dV$ be the time-dependent volume form over $\mathcal{M}$ corresponding to the density interpolant $\alpha_t$. Note that $\int_{\mathcal{M}} \hat{\alpha}_t = 1$. Moser's idea is to replace equation 2 with its continuous version:

$$\hat{\alpha}_0 = \Phi_t^* \hat{\alpha}_t, \quad t \in [0, 1] \tag{A1}$$

If equation A1 holds for all $t \in [0, 1]$ then plugging $t = 1$ leads to equation 2. Since equation A1 holds trivially for $t = 0$ (since $\Phi_0$ is the identity mapping), solving it amounts to asking that $\Phi_t^* \hat{\alpha}_t$ is constant, i.e.,

$$\frac{d}{dt} \Phi_t^* \hat{\alpha}_t = 0. \tag{A2}$$

The time derivative of $\Phi_t^* \hat{\alpha}_t$ can be computed with the help of the Lie derivative (e.g., Proposition 5.2 in Lang (2012)): If $\Phi_t$ is the flow corresponding to the time dependent vector field $v_t$ (see equation 3), and $\omega$ is a differential form then

$$\frac{d}{dt}(\Phi_t^* \omega) = \Phi_t^*(\mathfrak{L}_{v_t} \omega),$$

where $\mathfrak{L}$ denotes the Lie derivative. The Lie derivative $\mathfrak{L}_v \omega$ of a smooth vector field $v$ and smooth differential form $\omega$ can be computed using Cartan's "magic formula" (see e.g., Theorem 14.35 in Lee (2013)):

$$\mathfrak{L}_v \omega = i_v(d\omega) + d(i_v \omega),$$

where $i_v \omega$ is the interior multiplication of a vector field and a differential form defined by $(i_v \omega)(v_2, \ldots, v_n) = \omega(v, v_2, \ldots, v_n)$. In case $\omega$ is an $n$-form (as $\hat{\alpha}_t$ in our case) we have $d\omega = 0$ so the first term in the r.h.s. above vanishes. Lastly, we will need the following "trick":

$$\frac{d}{dt}(\Phi_t^* \hat{\alpha}_t) = \frac{d}{ds}\Big|_{s=t}(\Phi_s^* \hat{\alpha}_t) + \frac{d}{ds}\Big|_{s=t}(\Phi_t^* \hat{\alpha}_s).$$

Putting the last three equations together we get:

$$\frac{d}{dt}(\Phi_t^* \hat{\alpha}_t) = \Phi_t^*(\mathfrak{L}_{v_t} \hat{\alpha}_t) + \Phi_t^*\left(\frac{d}{dt} \hat{\alpha}_t\right) = \Phi_t^*\left(d(i_{v_t} \hat{\alpha}_t) + \frac{d}{dt} \hat{\alpha}_t\right). \tag{A3}$$

The theorem is proven if one can show that $v_t \in \mathfrak{X}(\mathcal{M})$ exists such that $d(i_{v_t} \hat{\alpha}_t) + \frac{d}{dt} \hat{\alpha}_t = 0$. The divergence operator is defined by the equality $d(i_w dV) = \text{div}(w)dV$, for a vector field $w \in \mathfrak{X}(\mathcal{M})$. Therefore $d(i_{v_t} \hat{\alpha}_t) = \text{div}(\alpha_t v_t)dV$. Denote $\hat{\gamma}_t = \frac{d}{dt} \hat{\alpha}_t$. Then we need to show that $v_t \in \mathcal{M}$ exists such that

$$d(i_{v_t} \hat{\alpha}_t) + \hat{\gamma}_t = 0. \tag{A4}$$

By the Hodge decomposition (see Theorem 4.18 in Morita (2001)) $\hat{\gamma}_t$ can be written as a sum of an exact and harmonic forms: $\hat{\gamma}_t = d\hat{\beta}_t + \hat{h}_t$. Since every harmonic form on a connected, compact,

oriented Riemannian manifold is a constant multiple of the Riemannian volume form, $cdV$ (see Corollary 4.14 in Morita (2001)), we have

$$0 = \frac{d}{dt}1 = \frac{d}{dt}\int_{\mathcal{M}}\hat{\alpha}_t = \int_{\mathcal{M}}\hat{\gamma}_t = \int_{\mathcal{M}}d\hat{\beta}_t + \int_{\mathcal{M}}\hat{h}_t = \int_{\mathcal{M}}\hat{h}_t = c\int_{\mathcal{M}}dV,$$

where in the second from the right equality we used Stokes Theorem (see e.g., Theorem 16.11 in Lee (2013)) and the fact that $\mathcal{M}$ has no boundary. This implies that $c = 0$, and

$$\hat{\gamma}_t = d\hat{\beta}_t. \tag{A5}$$

Using the correspondence between vector fields and $d - 1$ forms we let $\beta_t = i_{u_t}dV$, where $u_t \in \mathfrak{X}(\mathcal{M})$, and $d\beta_t = d(i_{u_t}dV) = \text{div}(u_t)dV$.

Lastly, consider $v_t$ defined as follows:

$$v_t = -\frac{u_t}{\alpha_t}. \tag{A6}$$

With this choice equation A4 is satisfied:

$$d(i_{v_t}\hat{\alpha}_t) + \hat{\gamma}_t = -d(i_{\frac{u_t}{\alpha_t}}(\alpha_t dV)) + i_{u_t}dV = 0.$$

The theorem is proven. $\square$

One comment is that for practically finding $v_t$, according to equation A6, we need to get $u_t$, which amounts to solving the Hodge decomposition equation, $\text{div}(u_t)dV = \hat{\gamma}_t$, that is equivalent to the following PDE on the manifold $\mathcal{M}$:

$$\text{div}(u_t) = \frac{d}{dt}\alpha_t. \tag{A7}$$

*Proof of Lemma 1.* The proof uses Stokes theorem:

$$\int_{\mathcal{M}}\text{div}(u)dV = \int_{\mathcal{M}}d(i_udV) = \int_{\partial\mathcal{M}}i_udV = 0,$$

where the last equality is due to the fact that either $\partial\mathcal{M} = \emptyset$, or, for $x \in \partial\mathcal{M}$, we have that $u(x) \in T_x\partial\mathcal{M}$, and therefore $(i_udV)(v_1, \ldots, v_{n-1}) = dV(u, v_1, \ldots, v_{n-1}) = 0$, for all $v_1, \ldots, v_{n-1} \in T_x\partial\mathcal{M}$. This implies $i_udV = 0$. $\square$

# B  Other proofs

*Proof of Theorem 2.* As we showed in the paper, our loss can be equivalently presented (up to constant factors) as

$$l(\theta) = D(\mu, \bar{\mu}_+) + (\lambda - 1)\int_{\mathcal{M}}\bar{\mu}_- dV$$

Where the first term $D(\mu, \bar{\mu}_+)$ is the generalized KL divergence which is non-negative and equals zero iff $\bar{\mu}_+ = \mu$ and since $\lambda \geq 1$ the second term is also non-negative and equals zero iff $\mu_- = 0$ or $\lambda = 1$.
First we show that $\bar{\mu} = \mu$ is a minimizer of the loss. Since we assumed $\mu \geq \epsilon$ we have that $\bar{\mu}_+ = \max(\mu, \epsilon) = \mu$ and $\bar{\mu}_- = \bar{\mu}_+ - \bar{\mu} = 0$. So both $D(\mu, \bar{\mu}_+)$ and $\int_{\mathcal{M}}\bar{\mu}_- dV$ are minimized, which means the entire loss is minimized.
Now lets assume $\bar{\mu}$ is a minimizer of the loss. If $\lambda > 1$ $\bar{\mu}$ has to minimize both terms, as we know there exists a minimizer that minimizes both of them. In particular for any $\lambda \geq 1$ we have that $\bar{\mu}$ minimizes $D(\mu, \bar{\mu}_+)$ meaning $\bar{\mu}_+ = \mu$. Now we have that $0 = 1 - 1 = \int_{\mathcal{M}}\bar{\mu}dV - \int_{\mathcal{M}}\mu dV = \int_{\mathcal{M}}\bar{\mu}_+ dV + \int_{\mathcal{M}}\bar{\mu}_- dV - \int_{\mathcal{M}}\mu dV = \int_{\mathcal{M}}\bar{\mu}_- dV$. So we get that $\mu_- = 0$. Finally $\bar{\mu} = \bar{\mu}_+ + \bar{\mu}_- = \mu + 0 = \mu$. $\square$

*Proof of Lemma 2.* Proposition 1.2 in Lang (2012) and Definition 1 in Section 4-4 in Do Carmo (2016) imply that for submanifolds with induced metric the Riemannian covariant derivative at $x \in \mathcal{M}$ satisfies $\nabla_{e_i}u = P_x\frac{\partial u}{\partial e_i}$, where $P_x$ is the projection matrix on $T_x\mathcal{M}$ introduced above.

Then, denoting $\boldsymbol{e}_1, \ldots, \boldsymbol{e}_n, \boldsymbol{n}_1, \ldots, \boldsymbol{n}_k$ an orthonormal basis of $\mathbb{R}^d$ where the first $n$ vectors span $T_{\boldsymbol{x}}\mathcal{M}$ and the latter $k$ span $N_{\boldsymbol{x}}\mathcal{M}$:

$$\mathrm{div}(\boldsymbol{u}) = \sum_{i=1}^{n} \langle \nabla_{\boldsymbol{e}_i} \boldsymbol{u}, \boldsymbol{e}_i \rangle_g = \sum_{i=1}^{n} \left\langle \boldsymbol{P}_{\boldsymbol{x}} \frac{\partial \boldsymbol{u}}{\partial \boldsymbol{e}_i}, \boldsymbol{e}_i \right\rangle = \sum_{i=1}^{n} \left\langle \frac{\partial \boldsymbol{u}}{\partial \boldsymbol{e}_i}, \boldsymbol{P}_{\boldsymbol{x}} \boldsymbol{e}_i \right\rangle = \sum_{i=1}^{n} \left\langle \frac{\partial \boldsymbol{u}}{\partial \boldsymbol{e}_i}, \boldsymbol{e}_i \right\rangle$$

$$= \sum_{i=1}^{n} \left\langle \frac{\partial \boldsymbol{u}}{\partial \boldsymbol{e}_i}, \boldsymbol{e}_i \right\rangle + \sum_{j=1}^{k} \left\langle \frac{\partial \boldsymbol{u}}{\partial \boldsymbol{n}_j}, \boldsymbol{n}_j \right\rangle = \mathrm{div}_E(\boldsymbol{u}),$$

$\square$

*Proof of Theorem 3.* From Theorem 6.24 in Lee (2013) there exists a neighbourhood $\Omega \subset \mathbb{R}^d$ of $\mathcal{M}$ such that the projection $\pi : \Omega \to \mathcal{M}$ is smooth over $\bar{\Omega}$ (i.e., can be extended to a smooth function over a neighborhood of $\bar{\Omega}$). Since $\mathcal{M}$ is compact, $\bar{\Omega}$ is also compact. According to Theorem 1 there exists a vector field $\boldsymbol{u}^\star \in \mathfrak{X}(\mathcal{M})$ so that $\mu = \nu - \mathrm{div}(\boldsymbol{u}^\star)$. We extend $\boldsymbol{u}^\star$ to $\bar{\Omega}$ by setting $\boldsymbol{u}^\star(\boldsymbol{x}) = \boldsymbol{u}^\star(\pi(\boldsymbol{x}))$, for $\boldsymbol{x} \notin \mathcal{M}$. Note that for $\boldsymbol{x} \in \mathcal{M}$ this definition coincides with the former $\boldsymbol{u}^\star$ defined over $\mathcal{M}$. Similarly to equation 18 we have that $\boldsymbol{u}^\star(\boldsymbol{x}) = \boldsymbol{P}_{\pi(\boldsymbol{x})}\boldsymbol{u}^\star(\pi(\boldsymbol{x}))$.

Corollary 3.4 in Hornik et al. (1990) shows that given a target smooth function $f : \bar{\Omega} \to \mathbb{R}$ and $\epsilon > 0$, there exists an MLP with $l$-finite smooth activation that uniformly approximate the first $0 \leq m \leq l$ derivatives of $f$ over $\bar{\Omega}$ with error at most $\epsilon$. An activation $\sigma : \mathbb{R} \to \mathbb{R}$ is $l$-finite if it is $l$-times continuously differentiable and satisfies $0 < \int_{-\infty}^{\infty} |\sigma^{(l)}| < \infty$. Note that sigmoid and $\tanh$ are $l$-finite for all $l \geq 1$, and Softplus is $l$-finite for $l \geq 2$.

Using this approximation result (adapted to vector valued MLP) there exists an MLP $\boldsymbol{v}_\theta : \mathbb{R}^d \to \mathbb{R}^d$ such that each coordinate of $\boldsymbol{u}^\star$ and $\boldsymbol{v}_\theta$ are $\epsilon$ close in value and first partial derivatives over $\bar{\Omega}$.

Now for arbitrary $\boldsymbol{x} \in \mathcal{M}$ we have

$$\bar{\mu}(\boldsymbol{x}) = \nu(\boldsymbol{x}) - \mathrm{div}_E(\boldsymbol{P}_{\pi(\boldsymbol{x})}\boldsymbol{v}_\theta(\pi(\boldsymbol{x})))$$

$$= \nu(\boldsymbol{x}) - \mathrm{div}_E\Big(\boldsymbol{P}_{\pi(\boldsymbol{x})}\boldsymbol{v}_\theta(\pi(\boldsymbol{x})) - \boldsymbol{P}_{\pi(\boldsymbol{x})}\boldsymbol{u}^\star(\pi(\boldsymbol{x}))\Big) - \mathrm{div}(\boldsymbol{u}^\star(\boldsymbol{x}))$$

$$= \mu(\boldsymbol{x}) - \mathrm{div}_E\Big(\boldsymbol{P}_{\pi(\boldsymbol{x})}\left[\boldsymbol{v}_\theta(\pi(\boldsymbol{x})) - \boldsymbol{u}^\star(\pi(\boldsymbol{x}))\right]\Big)$$

$$= \mu(\boldsymbol{x}) - \mathrm{div}_E\Big(\boldsymbol{P}_{\pi(\boldsymbol{x})}\boldsymbol{e}(\boldsymbol{x})\Big),$$

where we denote $\boldsymbol{e}(\boldsymbol{x}) = \boldsymbol{v}_\theta(\pi(\boldsymbol{x})) - \boldsymbol{u}^\star(\pi(\boldsymbol{x}))$. We will finish the proof by showing that

$$\left| \mathrm{div}_E\Big(\boldsymbol{P}_{\pi(\boldsymbol{x})}\boldsymbol{e}(\boldsymbol{x})\Big) \right| < c\epsilon$$

for some constant $c > 0$ depending only on $\mathcal{M}$. Note that the l.h.s. of this equation is a sum of terms of the form $\frac{\partial}{\partial x^i}\left((\boldsymbol{P}_{\pi(\boldsymbol{x})})_{i,j}\boldsymbol{e}(\boldsymbol{x})_j\right)$, where $(\boldsymbol{P}_{\pi(\boldsymbol{x})})_{i,j}$ is the $(i,j)$-th entry of the matrix $\boldsymbol{P}_{\pi(\boldsymbol{x})}$ and $\boldsymbol{e}(\boldsymbol{x})_j$ is the $j$-th entry of $\boldsymbol{e}(\boldsymbol{x})$. Since the value and first partial derivatives of $\pi$ and $\boldsymbol{P}$ (as the differential of $\pi$) over $\mathcal{M}$ can be bounded, depending only on $\mathcal{M}$, the theorem is proved.

$\square$

## C   Laplacian eigen function calculation

Given a triangular surface mesh $\mathcal{M}'$, we wish to calculate the $k$-th eigenfunction of the (discrete) Laplace-Beltrami operator over $\mathcal{M}'$. We will use the standard (cotangent) discretization of the Laplacian over meshes (Botsch et al., 2010). That is, we define $\boldsymbol{L}$ to be the cotangent-Laplacian matrix of the graph defined by $\mathcal{M}'$, and $\boldsymbol{M}$ the mass matrix of $\mathcal{M}'$, i.e., a diagonal matrix where $\boldsymbol{M}_{ii}$ is the area of the the Voroni cell of the $i$-th vertex in the mesh. We then calculate the eigenfunctions as the solution to the generalized eigenvalue problem $\boldsymbol{L}\boldsymbol{x} = \lambda_k \boldsymbol{M}\boldsymbol{x}$ where $\lambda_k$ is the $k$-th eigenvalue. We sample these $\mathcal{M}'$ piecewise-linear functions at centroids of faces.

## D   Linearization of the projection operator $\pi$

Since we only sample and derivate the projection operator $\pi : \mathbb{R}^d \to \mathcal{M}$ over $\mathcal{M}$, implementing equation 18 does not require knowledge of the full projection $\pi$. Rather, it is enough to use its first

order expansion over $\mathcal{M}$. For $\boldsymbol{x}_0 \in \mathcal{M}$

$$\pi(\boldsymbol{x}) \approx \pi(\boldsymbol{x}_0) + \boldsymbol{P}_{\boldsymbol{x}_0}(\boldsymbol{x} - \boldsymbol{x}_0) = \boldsymbol{x}_0 + \boldsymbol{P}_{\boldsymbol{x}_0}(\boldsymbol{x} - \boldsymbol{x}_0) = \hat{\pi}(\boldsymbol{x}_0, \boldsymbol{x}).$$

Now since $\pi(\cdot)$ and $\hat{\pi}(\boldsymbol{x}_0, \cdot)$ have the same value and first partial derivatives at $\boldsymbol{x}_0$ we can replace equation 18 for each sample point $\boldsymbol{x}_0 \in \mathcal{X} \cup \mathcal{Y}$, with

$$\boldsymbol{u}(\boldsymbol{x}) = \boldsymbol{P}_{\hat{\pi}(\boldsymbol{x}_0, \boldsymbol{x})} \boldsymbol{v}_\theta(\hat{\pi}(\boldsymbol{x}_0, \boldsymbol{x})).$$

## E   Unnormalized densities

As described in section 4, our formulation of the loss is dependent on knowing the volume of the manifold $\mathcal{M}$. For simple cases like the flat torus or the sphere, we have a closed form formula for this volume. For more general cases, we can show that we don't actually require to know this value, since we can work with unnormalized density functions:

$$\ell(\theta) = -\frac{1}{m} \sum_{i=1}^{m} \log \max \{\epsilon, \nu(\boldsymbol{x}_i) - \mathrm{div}_E \boldsymbol{u}(\boldsymbol{x}_i)\}$$

$$+ \frac{V(\mathcal{M})\lambda_-}{l} \sum_{j=1}^{l} \Big( \epsilon - \min \{\epsilon, \nu(\boldsymbol{y}_j) - \mathrm{div}_E \boldsymbol{u}(\boldsymbol{y}_j)\} \Big),$$

$$= \log V(\mathcal{M}) - \frac{1}{m} \sum_{i=1}^{m} \log \max \{\epsilon', \nu'(\boldsymbol{x}_i) - \mathrm{div}_E \boldsymbol{u}'(\boldsymbol{x}_i)\}$$

$$+ \frac{\lambda_-}{l} \sum_{j=1}^{l} \Big( \epsilon' - \min \{\epsilon', \nu'(\boldsymbol{y}_j) - \mathrm{div}_E \boldsymbol{u}'(\boldsymbol{y}_j)\} \Big),$$

where $\nu' = V(\mathcal{M})\nu \equiv 1$, $\boldsymbol{u}' = V(\mathcal{M})\boldsymbol{u}$, $\epsilon' = V(\mathcal{M})\epsilon'$, and $\log V(\mathcal{M})$ is a constant. Lastly note that the definition of $\boldsymbol{v}_t$ is invariant to this scaling and can be computed with the unnormalized quantities.

## F   Additional Experimental Details

We used an internal academic cluster with NVIDIA Quadro RTX 6000 GPUs. Every run and seed configuration required 1 GPU. All other experimental details are mentioned in the main paper. Our codebase, implemented in PyTorch, is attached in the supplementary materials. We will open-source it post the review process.

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

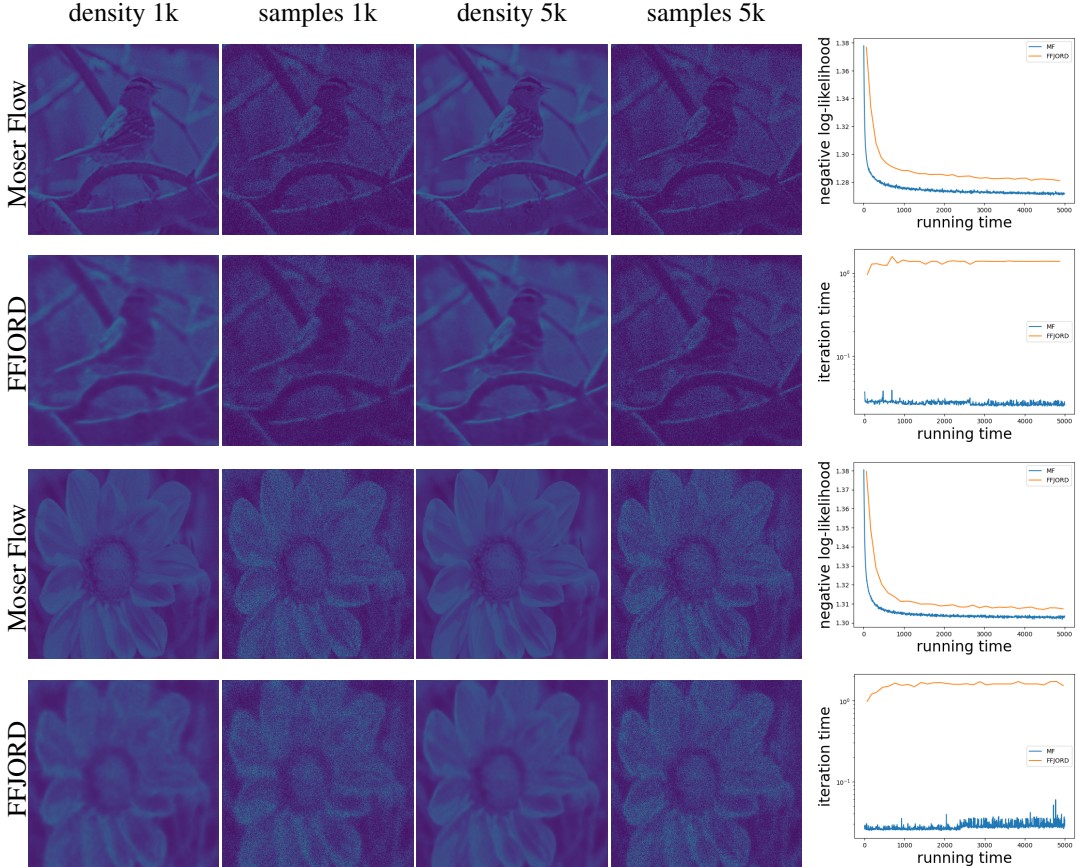

Figure A1: Comparing learned density and generated samples with MF and FFJORD at different times (in k-sec); top right shows NLL scores for both MF and FFJORD at different times; bottom right shows time per iteration (in log-scale, sec) as a function of total running time (in sec); FFJORD iterations take longer as training progresses. Flickr images (license CC BY 2.0): Bird by Flickr user "lakeworth" https://www.flickr.com/photos/lakeworth/46657879995/; Flower by Flickr user "daiyaan.db" https://www.flickr.com/photos/daiyaandb/23279986094/.