# OpenReview forum: "Moser Flow: Divergence-based Generative Modeling on Manifolds"
_NeurIPS.cc/2021/Conference — NeurIPS 2021 Oral_

### Official Review · Reviewer_96Cx · 2021-07-08

**Rating:** 7
**Confidence:** 3

**Summary:**

The paper introduces a new type of continuous normalizing flow (CNF) for manifold-valued data. Unlike in previous works, this Moser Flow (MF) models a density as a base density minus the divergence of a neural network. The authors prove that under certain assumptions this defines a probability distribution over the manifold. This approach has a different trade-off between inference and generation from other approaches: while the model density can be evaluated very cheaply, sampling from this model requires an ODE solver. The approach is qualitatively demonstrated on toy datasets on tori and the Stanford bunny, and quantitatively evaluated on a few datasets on 2-spheres.

**Ethical Concerns:**

I have no ethical concerns with this paper.

**Limitations And Societal Impact:**

I agree with the author's assessment that this work shares the same fundamental risks of all generative models. I do think that for the scope of this paper this acknowledgement is enough.

**Main Review:**

Modeling probability densities on manifolds is an important problem and Moser Flows present a new approach to solving it. Compared to existing methods, the MF setup presents a substantially different trade-off between the efficiency of likelihood evaluations and the efficiency of data generation. They may be thus be very relevant, in particular in applications where likelihood evaluations are more important than sampling.

Maybe the most important contributions of the paper are proposing this method, proving that it defines a probability distribution over the manifold, and proving that it presents a universal density estimator. These theoretical aspects seem correct to me, though I have not checked all the proofs. In some parts (for instance the proof of theorem 3) I would have appreciated more details and sometimes more precision, perhaps by providing an extended version of the argument in the supplementary material.

My main gripe with the method part of the paper is a certain lack of discussion. In particular:
1. What happens if the trained model still has $\bar{\mu}(x) < 0$ somewhere? Can the model still be useful to define a probability density? What are the limitations of the proposed regularization scheme (for instance when going to higher dimensions)?
2. How do Moser Flows scale? The authors briefly allude to the poor scaling in one sentence in the discussion, but this really deserves more space. Would the approach for instance work well in low-dimensional manifolds embedded in a high-dimensional space (like the hypothesized image manifold)?
3. A more in-depth comparison of MF to established flows on manifolds would really improve this paper.
4. Some part of the paper focuses on general Riemannian manifolds, while later the paper focuses on the case of submanifolds of a Euclidean space. How useful are Moser flows in the case when manifolds are not defined as submanifolds of some Euclidean space?
5. What if the manifold is not known a priori but you only have access to samples from it? Is there a way to extend this method such that an atlas for the manifold is learned as well?
I understand that space is at a premium at NeurIPS, but perhaps the author can make room to discuss some of these aspects by e.g. moving the proof of Thm. 3 into the appendix?

In the empirical part of the paper, the authors demonstrate that the idea works, at least in low-dimensional examples. The Stanford bunny results are cool to see. The quantitative evaluation is pretty thin, more metrics and comparisons to other baselines would have been better. I would also have liked to see an empirical verification that the model densities form a properly normalized probability density over the manifolds / that $\bar{\mu}(x)$ is positive everywhere. It is nice to see the speed-up in inference time in action, but it would have been even better to also include results on the sampling speed.

The paper is well written and clearly structured. Perhaps the presentation could have been further improved by hiding away more technical details in the supplementary materials and using the space for more detailed discussions and explanations (see above). Unlike many ML papers, it has properly sized labels in the figures (except, inexplicably, for the bottom right panel in Fig 3). The extensive supplementary materials and the provided code (clean, but not very documented) are good.

All in all, the authors present a novel solution to an interesting, relevant problem. The theory part of the paper is solid. While the empirical demonstration is thin, I do think that this paper has a lot of merit even without the usual benchmark table with N baselines in which the "Ours" entry has all the bold numbers. However, I do think that it currently lacks in-depth discussion of the properties of Moser Flows relative to previous approaches as well as on scaling it to higher-dimensional problems. With an extended discussion, I think this could make a good NeurIPS paper.

Minor comments and typos:
- Eq. (5): should this be a capital $\Phi$?
- Sec. 3.2: would be nice to explicitly describe sampling from the MF here somewhere
- line 140: "Stokes' theorem" with apostrophe, I believe
- around Eq. 14: more discussion of the choice of $\lambda$ would be great
- line 175: typo: superfluous "is"
- line 182: is the $x \in \mathcal{M}$ condition necessary here?
- Fig. 2: it would be great if the colour maps were scaled such that the empirical densities from the samples (based on histograms) and the density plot match
- Thm. 4: the statement "we conclude this section by proving" sounds a bit weird because there is no proof here (it is in the supplementary materials)
- line 220/221 read weird, typo?
- Fig. 3: how do FFJORD and MF compare in terms of log likelihood?
- Fig. 3: is this not a weird example to put on a torus b/c of the discontinuity at the identified edges of the images?
- Sec. 5.4: typo in the title, should be "Curved surfaces"
- Sec. 6: how does MF compare to these methods? This was superficially discussed before, but it would be great to re-visit in more detail here


**Time Spent Reviewing:**

5

---

> ### Author Response · Authors · 2021-08-10
> **Authors response**
>
> We thank the reviewer for a detailed and thoughtful review. We will incorporate other suggestions raised by the reviewer regarding the presentation and other experimental details in the final version. Below we address the main concerns raised in this review.
>
> **Q: “In some parts (for instance the proof of theorem 3) I would have appreciated more details and sometimes more precision, perhaps by providing an extended version of the argument in the supplementary material”.** \
> A: We will clarify theorems 2 and 3 in addition to clarifying points raised by other reviewers. We would be happy to learn of other parts that require clarification, if the reviewer feels there are such.
>
> **Q:  “What happens if the trained model still has $\bar{\mu}_- > 0$ somewhere? Can the model still be useful to define a probability density?“** \
> A: Yes, the model is still useful. Regardless of whether $\bar{\mu}- > 0$, the vector field $v_t$ (Equation 11) defines a valid diffeomorphism $\Phi$ and a corresponding density $\Phi_* \nu$ (the prior density pushed forward by $\Phi$). The quantity $\int \bar{\mu}-$ bounds the difference between the density $\Phi_* \nu$ and $\bar{\mu}+$. We will explain this in the paper. Lastly, we will add an experiment that shows, for different values of $\lambda$, the learned $\bar{\mu}-$, the difference between the $\bar{\mu}+$ and the density $\Phi_* \nu$.
>
> **Q: “What are the limitations of the proposed regularization scheme (for instance when going to higher dimensions)?”, “How do Moser Flows scale? The authors briefly allude to the poor scaling in one sentence in the discussion, but this really deserves more space.”**\
> A: This is a good question and indeed taking MF to high dimensions is a challenge which we mark as a future work. This challenge can be roughly broken to three: moving to log probabilities, estimating the integral of the negative density part (i.e., the regularization scheme), and computing the divergence of the vector field. We will expand the discussion of high dimension challenges in the paper to better facilitate future work. Regarding the particular regularization scheme question of this reviewer: One natural approach to scale it to high dimension is to note that Equation 12 implies that there is no negative part to $\bar{\mu}$ where $div(u)=0$. Therefore one option for high dimensions is to consider a concentrated prior $\nu$ (e.g., smoothed data distribution), restrict $u$ to vanish where $\nu$ vanish, and use $\eta=\nu$ in the importance sampling (see the equation following Theorem 3).
>
> **Q: “Would the approach for instance work well in low-dimensional manifolds embedded in a high-dimensional space (like the hypothesized image manifold)? “**\
> A: For low dimensional manifolds (regardless of the embedding space’s dimension) the challenge would be to define and sample the prior distribution $\nu$ (which can be done by projection), and to compute the divergence of $u$, which can be done in the tangent (low dimensional) linear space. As long as computing an orthogonal basis to this tangent space is tractable, the divergence can be computed efficiently.
>
> **Q: “How useful are Moser flows in the case when manifolds are not defined as submanifolds of some Euclidean space?“**\
> A: As long as one can compute (or approximate) the divergence of $u$ and sample according to the prior distribution $\nu$, MF can be applied.
>
> **Q: “What if the manifold is not known a priori but you only have access to samples from it? Is there a way to extend this method such that an atlas for the manifold is learned as well?“**\
> A: Learning the manifold together with the flow is potentially possible and worthy of future work. One option is to learn an implicit neural representation together with MF’s $u$.
>
>
>
> **Q: ”Fig. 3: how do FFJORD and MF compare in terms of log likelihood?“**\
> A: We will add log-likelihood versus time graph to the paper. This result can be found here: https://sites.google.com/view/2021moserflowrebuttal/; these are updated results of our method showing MF outperforms FFJORD by a large margin.

---

> > ### Comment · Reviewer_96Cx · 2021-08-17
> > **Reply to author response**
> >
> > Thanks for answering most of my questions. I'm happy with the responses and confident that with the promised changes, in particular the extended discussion, this will make a good NeurIPS paper. I changed my score to reflect this.

---

### Official Review · Reviewer_VH2c · 2021-07-18

**Rating:** 7
**Confidence:** 2

**Summary:**

The paper tackles the problem of learning flow-based generative models. The proposed approach, a new instance of continuous normalizing-flow methods, models the density as a difference between the prior and a divergence term, parameterized by a neural network.

The claims are that:
* MF constitutes a universal density approximator,
* MF is more efficient to train than alternative CNFs; this is demonstrated empirically and the intuition is that MF does not require invoking or backpropagating through an ODE during training (note however that solving an ODE is required for sampling),
* MF improves upon alternative CNFs in terms of density estimation, sample quality, and training complexity,
* MF demonstrates for the first time the use of flow models for sampling from general curved surfaces.

**Limitations And Societal Impact:**

The authors discuss the potential negative societal impact in the Discussion and limitations section.

**Main Review:**

Learning flow-based generative models is an active research area. The current paper designs a new model that aims to improve upon recently published continuous flow generative models (Riemannian CNFs, FFJORD). From this perspective, the approach is well-timed, theoretically motivated, and relevant to the community. I am not familiar with a similar approach, but note that I am not familiar with all the literature on continuous flow generative models.

*Experiments*:

My main concerns (questions) regard the empirical claims:
* Figure 2: The experiment is a great illustration of the proposed method, demonstrating that it can model complex (multi-modal, discontinuous) datasets. That said, prior methods, e.g. FFJORD, have also demonstrated that they can model these distributions (by warping an isotropic Gaussian), so this result is not specific to the current method.
* The claims made regarding Figure 3 are that 1) the proposed method is computationally more efficient than FFJORD and 2) MF captures high-frequency details better. Unfortunately, the claims are illustrated on a single image, without a numerical evaluation. Have you considered performing this evaluation across several images (e.g. the MNIST, Omniglot datasets) and reporting aggregated results (similar to the FFJORD paper)?
* Figure 5 illustrates the approach on the Stanford Bunny surface. Unfortunately, it’s hard to evaluate the difficulty of the problem given no baseline approaches. Since this is a major claim in the abstract, could you discuss why none of the previous approaches can be applied to this setting for comparison?
* Table 1: The most significant result is that the proposed approach outperforms prior methods from “Riemannian Continuous Normalizing Flows”. Since the prior results are extracted from the cited paper, and the methods there chosen to have approximately the same number of parameters, I think the results would be stronger if the authors discussed whether there is a significant difference in parameter size.

**Time Spent Reviewing:**

4-6

---

> ### Author Response · Authors · 2021-08-10
> **Authors response**
>
> We thank the reviewer for a detailed and thoughtful review. We will incorporate other suggestions raised by the reviewer regarding the presentation and other experimental details in the final version. Below we address the main concerns raised in this review.
>
> **Q: The claims made regarding Figure 3 are illustrated on a single image, without a numerical evaluation.** \
> A: The image we used in Figure 3 (i.e., Camera-man) was chosen as a standard computer vision test bed challenge image with interesting structure and high frequencies (e.g, camera stand, hand over coat, building in the background, etc.). Datasets like MNIST and Omniglot lack such level of intricate details; however, if the reviewer feel it's helpful we will add one or more images from such datasets. We will also add a log-likelihood versus time curve to this example showing Moser Flow (MF) outperforms FFJORD by a large margin; see https://sites.google.com/view/2021moserflowrebuttal/ .
>
> **Q: Figure 5 illustrates the approach on the Stanford Bunny surface ... could you discuss why none of the previous approaches can be applied to this setting for comparison?** \
> A: Previous Continuous Normalizing Flows (CNFs) approaches suffer from key challenges that prohibit their use for arbitrary implicit surfaces; for example, they would require building a differentiable ODE solver over the implicit surface which is non-trivial. MF, in contrast, only requires approximating a differential operator over the implicit surface. One could potentially conceive mechanisms to adapt these approaches to arbitrary implicit surfaces such as the Stanford Bunny but we believe this adaptation is non trivial, worthy of its own research paper, and cannot be considered as existing baselines. Indeed, we are not aware of existing CNF approaches demonstrated on non-constant curvature surfaces, let alone highly curved surfaces such as the Stanford Bunny.
>
> **Q: Table 1: The most significant result is that the proposed approach outperforms prior methods from “Riemannian Continuous Normalizing Flows”. Since the prior results are extracted from the cited paper, and the methods there chosen to have approximately the same number of parameters, I think the results would be stronger if the authors discussed whether there is a significant difference in parameter size.** \
> A: As mentioned in Section 5.3 we used an MLP with 6 layers of 512 neurons which is indeed larger than the architecture used in Riemannian CNF paper. Note that there are three points to note in this regards: First, in Figure 3 (we also have an updated version of this figure with log-likelihood graph and fixed time evaluations in this link: https://sites.google.com/view/2021moserflowrebuttal/ ) we show a comparison of FFJORD and MF with the similar architecture (up to FFJORD being time dependent) and show the benefit of the latter. Second, Riemannian CNFs are not efficient due to the ODE solver and therefore are prohibitive for large architectures. For example Riemannian CNF with 3x64 MLP trains for approximately 10 hours on the Earth-Fire dataset, while MF with 6x512 MLP trains for approximately 2 hours. Third, the Riemannian CNFs are training a **time dependent** vector field in contrast to MF that has a fixed time dependency (i.e., trained without a time parameter). Therefore, the direct comparison of models based on simple parameter counts is not so evident.

---

> > ### Comment · Reviewer_VH2c · 2021-08-26
> > **Thank you for the response**
> >
> > Thank you for the thorough rebuttal, including the additional results, and I apologize for the slow response. The rebuttal addresses my questions & reading the other reviews as well, I am changing my score to recommend acceptance.

---

### Official Review · Reviewer_STbS · 2021-07-21

**Rating:** 8
**Confidence:** 3

**Summary:**

The authors define a continuous normalizing flow on an arbitrary Riemannian manifold through the divergence of a neural network using the Moser theorem. They motivate their approach and show how this flow can be represented and learned. By applying their method to a toy distributions and a real world earth and climate dataset, they demonstrate the effectiveness of the method and that they can outperform competing procedures. Furthermore, they show that their method is much faster and more efficient than FFJORD when learning a complex two dimensional density.

**Limitations And Societal Impact:**

Currently, the method cannot be applied to high dimensional manifolds. Although this is unfortunate, competing methods struggle with this problem as well.

**Main Review:**

The authors use a very creative approach to define a continuous normalizing flow through the divergence of a neural network. Usually, they are constructed as the solution of an ODE and solving it can be time consuming. Here, the authors use the Moser theorem and use that the divergence of a vector field can be the difference of two distributions, i.e the base distribution and a distribution approximating the target. The power of the method is that it can directly operate on an arbitrary Riemannian manifold without the requirement of an invertible map from $\mathrm{R}^d$ to the manifold. So far, only a small number of methods exist which are able to do that.
Implementing and learning the flow, however, requires some heuristic tricks and since I am not an expert on the Moser theorem and its implications in this context, I am not sure how valid they are.
Anyway, the method seems to be very efficient and outperforms competing approaches on small dimensional toy and real world datasets.

**Time Spent Reviewing:**

4

---

> ### Author Response · Authors · 2021-08-10
> **Authors response**
>
> Thank you for your review. (We could not identify a particular question for the rebuttal.)

---

### Official Review · Reviewer_oTmD · 2021-07-22

**Rating:** 8
**Confidence:** 4

**Summary:**

This paper presents Moser Flow a new class of Continuous Normalizing Flows for Riemannian Manifolds. The main approach exploits results in differential geometry (Moser 1965) to construct a vector field that satisfies the normalizing equation given by the pullback of the volume form under $\Phi$. The approach is both sensible and provides direct advantages over Riemmanian CNF's as training does not require solving the ODE using a solver. The authors also consider Moser Flows for Euclidean Submanifolds under the Induced metric and show that Moser Flows can learn arbitrary distributions on manifolds of interest (---i.e. compact, boundryless etc...). Finally, the experiments conducted are both interesting in their variety and illuminating in that they show the effectiveness of the proposed method. Overall, this paper represents a non-trivial step forward---manifold or otherwise---for Continuous Normalizing Flow research.

**Main Review:**

Overall, I really enjoyed reading this paper and I think for the most part it is exceptionally well written. The main ideas are clear but maybe the authors could do a bit more to contrast the difference between Riemannian CNF by Mathieu and Nickel 2020. I would also like to highlight, what I believe are the principal strengths of the approach. First, unlike conventional Riemannian CNF's----and CNF's more broadly---one does not need to solve the ODE using off the shelf solvers that may introduce additional sources of error. This i believe is a significant win. To this end I think the paper would be strengthened by considering similar datasets/experiment pipelines of Euclidean CNF's (beyond 2D toy data and Figure 3) to see if Moser Flows can handle conventional non-manifold specific generative modellng datasets which are significantly larger, both in dimensionality and size, than the ones considered here.

In terms of key weaknesses, I don't find that this paper has any major drawbacks. One could criticize that it leans too heavily on the original Moser's results (e.g. even the interpolant is borrowed) but I think this is folly. The adaptation of the method to Riemannian CNF's brings about a lot of technical novelty which should not be understated. In terms of experiments, I wish the authors also compared with concurrent NF on manifold baselines like Neural Ordinary Manifold Differential Equations by Lou et. al 2020 and Falorsi, L. and Forré 2020. These methods have subtle but important differences than Riemannian CNF by Mathieu and Nickel 2020. In terms of presentation, the supplementary material could be strengthened. Right now there is no explicit/formal proof of Theorem 2 and 3. I do concede that the main arguments for Theorem 3 are present in the main text, but this is not a formal proof.

Questions/Minor Comments:
- The divergence computation in Riemannian CNF's relied on the Hutchinson Trace Estimator, was this also used for Moser Flow?
- This maybe my misunderstanding, but I feel like there is a missing step in the proof of Lemma 2 as the final equality in RHS of the equation below line 31 does not equal the first equality of the line below. Unless, $\sum_{j=1}^k \langle \frac{\partial u}{\partial n_i}, n_i \rangle$ (Note that there is a typo with subscript $i$ in the summation over $k$,it should be over $j$) is $0$.




**Time Spent Reviewing:**

6

---

> ### Author Response · Authors · 2021-08-10
> **Authors response**
>
> We thank the reviewer for a detailed and thoughtful review. We will incorporate other suggestions raised by the reviewer regarding the presentation and other experimental details in the final version. Below we address the main concerns raised in this review.
>
> **Q: “maybe the authors could do a bit more to contrast the difference between Riemannian CNF by Mathieu and Nickel 2020”**\
> A: Both RCNF and MF represent a diffeomorphism on a manifold parameterized via a tangent vector field. While calculating the likelihood in RCNF requires solving a global ODE over the manifold, the likelihood computation with MF only requires local information in the form of differentiation (divergence operator). The flip side is that MF requires an extra regularization term (cancelling the negative part of the learned density). Still, the fact that MF avoids solving an ODE makes it a much simpler to train and computationally efficient, allowing it to work with more complicated geometries (e.g., Stanford Bunny) and larger/deeper networks than is possible with RCNF.
>
> **Q: “I wish the authors also compared with concurrent NF on manifold baselines like Neural Ordinary Manifold Differential Equations by Lou et. al 2020 and Falorsi, L. and Forré 2020.”**\
> A: Similarly to RCNF, both Lou et al. 2020 and Falsori and Forre 2020 suggest solving an ODE over the manifold to calculate the target density at data points. Furthermore, they also demonstrate solely on constant curvature manifolds: the sphere and the hyperbolic plane.
>
> **Q: “Right now there is no explicit/formal proof of Theorem 2 and 3. “**\
> A: We will add a formal proof of Theorem 3 to the supplementary. Theorem 2 is a direct corollary of theorem 1 and lemma 1, we will rename it to Corollary and will clarify it as-well.
>
> **Q: “The divergence computation in Riemannian CNF's relied on the Hutchinson Trace Estimator, was this also used for Moser Flow?”**\
> A:  No. We computed the divergence analytically due to the low dimensionality of the experiments.
>
> **Q: “I feel like there is a missing step in the proof of Lemma 2 as the final equality in RHS of the equation below line 31 does not equal the first equality of the line below. Unless, $\sum_{j=1}^k \langle \frac{\partial u}{\partial n_i},n_i\rangle$ is zero”**\
> A: Since $n_1,\ldots,n_k$ is a basis of $N_x\mathcal{M}$, and we assumed $u$ is constant in the normal directions to $\mathcal{M}$, this sum is indeed zero. We will fix the typo in the indexing.

---

> > ### Comment · Reviewer_oTmD · 2021-08-16
> > **Re:response**
> >
> > Thank you for your response to my questions. I believe they have answered all my concerns and I am more confident in maintaining my current assessment of the paper.

---

### Official Review · Reviewer_R6hn · 2021-08-03

**Rating:** 7
**Confidence:** 4

**Summary:**

This paper proposes an alternative way to train a continuous normalizing flow (CNF), inspired by the Dacorogna-Moser transport. The proposed training does not require numerical integration; instead, the likelihood of the flow can be directly evaluated by computing the divergence of the mass flow rate. The idea is tested on boundaryless manifolds, such as the flat torus for toy data and implicit surfaces.

**Limitations And Societal Impact:**

See above

**Main Review:**


*Reason for rating*: I think this work is quite novel and neat and complements the current flow literature well so I vote for acceptance. I would be happy to increase my rating if the author can address some of my concerns about scalability and the limitations that I mentioned below.


### Significance and novelty

The proposed method is very neat, and largely simplifies the computational cost of CNF during training time. The likelihood of the model can be computed exactly in on a single forward evaluation (modulo the potential need of trace estimation), as opposed to numerically integrating the underlying ODE like FFJORD. The proposed flow is also provably universal (despite taking a simpler form, as per eq. 11, rather than having a free form velocity field).

### Scope

The proposed method is mainly designed for modeling data on manifolds, which is empirically tested on a few toy and real datasets. More specifically, the manifolds need to be compact, boundaryless, and connected for the proposed method to work, which slightly limits its applicability and at the same time also leaves room for innovation.

### Theoretical soundness

I only check for the main development in section 3 (so I cannot speak for section 4), which looks correct to me, except for the conditions in the statement of theorem 3. I am not quite sure about "For $\lambda\geq1$ and small $\epsilon>0$". Does the choice of $\epsilon$ depend on $\mu$ or $\lambda$? Or does it need to be asymptotically small? Also, why does $\lambda$ need to be greater than 1 if it's only a regularizer? It might help a bit to discuss the necessity of these conditions.


### Limitation

- (Regularization vs reparameterization) Most normalizing flows induces a valid probability density by design (such as regular CNFs which leverage the Lipschitzness of regular neural network architecture, residual flows that reparameterize the weights by spectral normalization, autoregressive and coupling blocks with masks over weight parameters, etc). However, the Moser flow needs to regularize the "negative part" of the density. It has the following disadvantages.
    1. Moser flow is not guaranteed to induce a valid probability density.
    2. It requires tuning an additional hyperparameter for regularization, $\lambda$. Is it a potential cause for instability during training? I actually think some negative results on this might help people better understand the work and think about how to improve upon it. How is it chosen as a hyperparameter (i.e. based on what criteria)? The chosen values (2 and 100) seem a bit drastic, with no discussion provided.
    3. The regularization seems to work at least empirically across the experiments conducted in this paper, but I suspect it might not scale well to higher dimensional problems, especially if $\eta$ is uniform.

- Section 5.2: It's obvious that the proposed method is faster to train than FFJORD since it does not require integration (which the authors show in Fig 3). But given that the proposed flow has a simple time conditioning mechanism, as opposed to the free form conditioning that's allowed when using FFJORD, I think it'll be fairer and more meaningful to also compare the increase in likelihood in runtime and/or in #iterations. Also, since FFJORD in that experiment only uses a time-independent vector field, the expressivity of the baseline is also weakened.

### Additional questions

- Is the trace operator computed exactly or estimated? I imagine it is done exactly given the low dimensionality of the problems, but it might be better to mention the potential problem of scaling up and the need for approximation.
- Is it possible to generalize the flat torus for 2D toy data to accommodate higher dimensional problems lying in the Euclidean space? such as dequantized image data that live in a [0, 1]^D hypercube.
- typo? line 83 "from" instead of "form"?





**Time Spent Reviewing:**

5

---

> ### Author Response · Authors · 2021-08-10
> **Authors response**
>
> We thank the reviewer for a detailed and thoughtful review. We will incorporate other suggestions raised by the reviewer regarding the presentation and other experimental details in the final version. Below we address the main concerns raised in this review.
>
> **Q: “Does the choice of $\epsilon$ depend on $\mu$ or $\lambda$?”**\
> A: We will clarify the dependence of $\epsilon$ on $\mu$ in the paper. The dependence is as follows: Since $\mu_+\geq \epsilon$ by construction, the claim will be true if $min_{x\in\mathcal{M}} \mu(x) > \epsilon$. Since we assume that $\mu$ is a continuous density over a compact manifold, such an $\epsilon$ exists. In practice, an arbitrarily small choice of $\epsilon$ does not really affect the learned density.
>
> **Q: “Why does $\lambda$ need to be greater than 1 if it's only a regularizer?”**\
> A: As we described in Theorem 3, this condition is sufficient for assuring that the global minimum of the loss (in Equation 14) is the sought after density $\mu$. If this condition is not held, then we have no guarantee that the minimum of our loss will be $\mu$; please see the discussion prior to the formulation of Theorem 3.
>
>
> **Q:  “It requires tuning an additional hyperparameter for regularization $\lambda$. Is it a potential cause for instability during training? I actually think some negative results on this might help people better understand the work and think about how to improve upon it. How is it chosen as a hyperparameter (i.e. based on what criteria)? The chosen values (2 and 100) seem a bit drastic, with no discussion provided”**\
> A: Thank you for this question. As a general rule of thumb: as $\lambda$ is increased the negative part of the integral is decreased. Note that even if there is a non-negligible negative part to the learned density $\bar{\mu}- >0$ the vector field still generates a valid density, and the discrepancy between the generated density $\Phi_*\nu$ and $\bar{\mu}+$ can be analytically bounded by $\int_{\mathcal{M}}\bar{\mu}_-$. If the reviewer feels that’s helpful we will add an experiment showing the tradeoff between $\lambda$ and the negative part of $\bar{\mu}$.
>
> **Q: “The regularization seems to work at least empirically across the experiments conducted in this paper, but I suspect it might not scale well to higher dimensional problems, especially if η is uniform”.** \
> A: This is a good remark and indeed taking MF to high dimensions is a challenge which we mark as a future work and contains three parts: moving to log probabilities, estimating the integral of the negative density part, and computing the divergence of the vector field. As correctly noted by this reviewer, one part of the challenge is dealing with the regularization term (estimating the negative density part). We will expand the discussion of high dimension challenges in the paper to better facilitate future work. Regarding this particular part: One natural approach here is to note that Equation 12 implies that there is no negative part to $\bar{\mu}$ where $div(u)=0$. Therefore one option for high dimensions is to consider a concentrated prior $\nu$ (e.g., smoothed data distribution as in recent score-based generative models), restrict $u$ to vanish where $\nu$ vanish, and use $\eta=\nu$ in the importance sampling (see the equation following Theorem 3).
>
> **Q: “Section 5.2: I think it'll be fairer and more meaningful to also compare the increase in likelihood in runtime and/or in #iterations.”**\
> A: We will add this to the paper. The likelihood versus time result can be found here: https://sites.google.com/view/2021moserflowrebuttal/ ; these are updated results of our method showing MF outperforms FFJORD by a large margin.
>
> **Q: “Section 5.2: since FFJORD in that experiment only uses a time-independent vector field, the expressivity of the baseline is also weakened.”**\
> A: This is incorrect: FFJORD in fact uses time dependent vector fields.
>
> **Q: “Is the trace operator computed exactly or estimated?“**\
> A: We compute the divergence analytically in the examples in this paper. As noted above, this is indeed one part out of three required for taking MF to higher dimensions and will be elaborated in the next version of the paper. In particular, the divergence can be approximated using the Hutchinson estimator.
>
> **Q: “Is it possible to generalize the flat torus for 2D toy data to accommodate higher dimensional problems lying in the Euclidean space? such as dequantized image data that live in a [0, 1]^D hypercube”**\
> A: Yes, this seems doable. It means we will work on the D-dimensional torus.

---

> > ### Comment · Reviewer_R6hn · 2021-08-25
> > **thank you for the reply, score updated**
> >
> > I thank the reviewers for the reply and the new ablation. Although it has some limitations for scaling up to high dimensionality, it creates new avenues for research in the future (which I hope the authors will take the extra page to expand on should the paper get accepted). I believe this is good work, and I'd like to see it accepted to NeurIPS. The reviewers also addressed most of my concerns, so I updated my score to 7.

---

### Decision · Program_Chairs · 2021-09-27

**Decision:**

Accept (Oral)

**Comment:**

The following is a summary of the pros and cons that resulted from the reviews and the discussion period.

pros:
* novel and nontrivial contributions to the flow literature. (R6hn, oTmD, STbS, VH2c, 96Cx)
   - reduces computational cost to train CNFs by no longer needing to solve an ODE.
   - likelihood computation does not require evaluating an ODE, constrast to regular CNFs.
   - Probably universal density estimator under suitable assumptions.
* interesting variety of experiments that show the effectiveness of the proposed method. (oTmD)
* for the most part well written (oTmD, 96Cx)

cons:
* limitations:
    - scalability limitations to higher dimensional problems. --> authors respond that higher dimensions are indeed challenging and promis to expand on the precise challenges in an updated version of the paper. (R6hn, 96Cx)
    - regularization of the negative part of the density in moser flows introduces an additional hyperparameter for tuning. worries that the regularization parameter might lead to unstable training of chosen wrongly. --> authors added experiments to show the behavior of the model under different ranges of this hyperparameter. (R6hn)
   - can the negative density also lead to moser flow not producing a valid density? --> authors rebut this in their response. (R6hn, 96Cx)
* confusion around theorem 3, exact or approxiamte computation of traces etc. --> solved by author response.(R6hn)
* comparison with other normalizing flow methods on manifolds would strengthen the paper. --> addressed in discussion. (oTmD, 96Cx)
* introducing an explicit proof of theorem 2 and 3 would improve the paper (arguments for proof can now be found in main paper). --> addressed in discussion. (oTmD, 96Cx)
* Clarifications needed about applicability of method. Can Moser flows still be used for manifolds that are not submanifolds of a Euclidian space? What if the manifold is not known?  --> discussed in author response. (96Cx)
* improvements needed in terms of evaluation:
   - figure 2: prior methods can also model this distribution, so the success of the proposed method over previous methods is unclear. (VH2c)
   - quantitative evaluation would make the claims stronger of being more computationally efficient, capturing densities better (e.g. Fig 3). --> addressed in author response.  (VH2c, 96Cx)
   -  comparison is needed with FFJORD in terms of both likelihood and runtime/#iterations. --> the authors include a link in their response with these results. (R6hn)
   - figure 5: why can't previous approaches be used to model this problem? --> addressed in author response. (VH2c)
   - table 1: how do the number of parameters of this method compare to previous work? --> addressed in author response. (VH2c)

Authors and reviewers engaged into active discussions during the discussion period. Most of the concerns raised by the reviewers were addressed, leading to 3/5 reviewers raising their scores and a consensus to accept this submission. Considering the nontrivial contribution of this work and the future research it could lead to, the recommended decision is accept.